# FAACL: FEDERATED ADAPTIVE ASYMMETRIC CLUSTERED LEARNING

## ABSTRACT

Asymmetric clustering has remained an unexplored problem in Clustered Federated Learning (CFL), diverging from the traditional approach of forming independent, non-interacting clusters. Previous methodologies have been limited to either separating devices with different data quality into distinct clusters or merging all devices into a single cluster, both of which compromise either data utilization or model accuracy. We propose a new federated learning technique where some devices may contribute to the training of the models of other devices, but without enforcing reciprocity, leading to a form of asymmetric clustering. This is beneficial in a variety of situations including scenarios where it is desirable for a device with high quality data to help train the model of a device with low quality data, but not vice-versa. This method not only enhances data utilization across the devices, but also maintains the integrity of high-quality data. Through a rigorous theoretical analysis and empirical evaluations, we demonstrate that our approach can efficiently find high quality (asymmetric) clusterings for numerous devices, achieving competitive performance metrics on existing CFL benchmarks.

## 1 INTRODUCTION

Federated learning (McMahan et al., 2017) is a machine learning technique designed to train algorithms across decentralized devices while keeping data localized, thus addressing privacy, security, and data access challenges. Unlike traditional centralized machine learning methods where all data is uploaded to one server, federated learning allows for the model to be brought to the data source where training occurs. This approach is particularly valuable in scenarios where data privacy is important, such as in healthcare, finance, and mobile computing. For example, smartphones that utilize predictive text input features can improve their models using federated learning by learning from user interactions without ever needing to upload individual typing data to a central server. However, federated learning introduces complexities such as handling non-IID (independently and identically distributed) data across various devices, dealing with devices that have varying computational and storage capacities, and managing communication costs and efficiencies.

In practice, it is common for devices to encounter data from diverse distributions. Since heterogeneous data may induce different optimal predictors at different devices, this has led to the development of personalized federated learning techniques (Fallah et al., 2020). A popular approach consists of training a global predictor that is adapted or fine-tuned for each device. However, this assumes that the optimal predictors at each device are similar enough that fine-tuning / adapting a global predictor will be sufficient. In cases where some optimal predictors are very different and fine-tuning is insufficient, then clustered federated learning (Sattler et al., 2020; Mansour et al., 2020) becomes attractive. For instance, in mobile keyboard prediction, where users from different regions have distinct linguistic preferences and slang, fine-tuning a single global model may not be effective; clustered federated learning, on the other hand, allows for creating separate models for different linguistic groups to ensure that predictions remain relevant and accurate. In clustered FL, devices are partitioned in clusters such that devices share models only with the other devices in their cluster. When clusters combine devices with similar data while making sure that devices with very different data are in different clusters, then learning will be more effective. Existing techniques for clustered FL (Sattler et al., 2020) can learn clusters dynamically. However, most existing techniques assume a fixed number of clusters that is known a priori and all existing techniques assume that each device contributes to a single cluster. As we will explain later, this last assumption is suboptimal in asymmetric situations where training with the data of a device could help improve the prediction accuracy of other clusters

in addition to the cluster that this device belongs to. We describe a technique that relaxes those two assumptions.

Determining the correct number of clusters beforehand can be challenging. This is particularly true in federated learning environments where data is distributed across numerous devices with potentially diverse data distributions. Static clustering methods that assume a fixed number of clusters can lead to inefficiencies and inaccuracies, as they might not accommodate the dynamic nature of real-world data, which can vary in terms of volume, variety across different devices.

Adaptive clustering addresses this limitation by employing algorithms that dynamically adjust the number of clusters based on the evolving characteristics of the data. Instead of pre-defining a cluster count, adaptive clustering methods continuously analyze the incoming data and modify the cluster count in real-time. This flexibility allows the learning process to maintain high levels of efficiency and adaptability.

In federated learning, asymmetric scenarios often arise where the benefits of model sharing are not reciprocal between devices. For instance, consider a situation involving two devices, device $A$ and device $B$. Device $A$ has a large dataset characterized by the underlying conditional distribution $p_A(y \mid x)$, whereas Device $B$ has a smaller dataset with a similar conditional distribution $p_B(y \mid x)$ that matches $p_A(y \mid x)$ for $90\%$ of the inputs $x$. For device $A$, incorporating device $B$'s data could potentially introduce a bias that might degrade the accuracy of its own model because of the $10\%$ divergence in their data distributions. Thus device $A$ would not wish to train on data from device $B$. On the other hand, for device $B$, clustering with device $A$ could significantly reduce variance owing to the greater volume of data it would benefit from, thereby enhancing its overall performance (reduction in variance outweighs the bias introduced). For example, regarding keyboard prediction, smartphone users of a rare dialect may benefit from the model trained with a large user base of a similar common dialect, but not vice-versa.

The key contributions of our research are outlined as follows:

- **Introduction of Asymmetric Clustering**: We propose the novel concept of asymmetric clustering, enabling a more flexible and dynamic cluster formation that better reflects the diversity of data quality and distribution among devices.
- **Development of FAACL**: We implement asymmetric clustering within the framework of clustered federated learning, which not only groups non-reciprocal devices into distinct clusters but also establishes inter-cluster relationships.
- **Integration of statistical tests and bounds**: We incorporate a robust statistical test, the Wilcoxon signed rank test (Wilcoxon, 1992) and Hoeffding's bound (Hoeffding, 1994) to guide the cluster formation process.
- **Empirical Validation**: Through extensive experiments, we demonstrate FAACL's competitive performance compared to traditional baselines, and its effectiveness and scalability in diverse federated environments.

The paper is structured as follows. Section 2 provides some background about Clustered Federated Learning. Section 3 describes related work in clustered federated learning. Section 4 describes the proposed technique FAACL. Section 5 demonstrates FAACL empirically on some benchmarks. Finally, Section 6 concludes our work.

## 2 BACKGROUND AND NOTATION

Consider a set of $n$ devices denoted as $\mathcal{D}=\{d_1, ..., d_n\}$. For each device $d$ in this set, we denote its associated dataset as $Z_d$. Each data point within this dataset, represented as $z=(x,y)$, is assumed to be sampled from an underlying distribution, which we denote as $P_d(z)$, where $z \in Z_d$. Additionally, we partition the dataset $Z_d$ for each device into three subsets: the training set $Z_d^{train}$, the validation set $Z_d^{val}$, and the test set $Z_d^{test}$.

In Federated learning, the goal is to train predictors in a distributed way without the data leaving each device. Consider the loss function $\ell(\theta, z)$ of the model parameterized by $\theta$ on data point $z = (x, y)$. In global federated learning, the objective objective is to train a global model $\theta$ by minimize the population loss $L(\theta)$ represented as follows.

$$L(\theta) = \sum_{d \in \mathcal{D}} E_{z \sim P_d(z)}[\ell(\theta, z)] \tag{1}$$

For example, in FedAvg (McMahan et al., 2017), a global model $\theta$ is trained in a distributed way by computing local gradients of the loss function at each device ($\delta_d \leftarrow \sum_{z \in Z_d^{train}} \nabla \ell(\theta, z)$), which

are then sent to a server that aggregates them ($\delta \leftarrow \sum_{d \in \mathcal{D}} \frac{|Z_d^{train}|\delta_d}{\sum_{d \in \mathcal{D}}|Z_d^{train}|}$) before returning them to the devices that each update their copy of the global model ($\theta \leftarrow \theta - \delta$). This approach (as well as other global FL techniques (Zhang et al., 2021)) work well when the data at each device comes from similar distributions (i.e., homogeneous case). Personalized federated learning can deal with a small degree of data heterogeneity by tuning the global model into personalized models (Kulkarni et al., 2020).

In the case of high heterogeneity, clustered federated learning (Sattler et al., 2020; Ghosh et al., 2020) clusters devices with similar data distributions and learns a separate model for each cluster. We define $C_j$ as the $j^{th}$ cluster. Each cluster $C_j$ consists of two primary components:

- Component 1: A set of devices, $C_j.D$.
- Component 2: A cluster model parameterized by $C_j.\theta$.

We define a clustering, denoted as $\mathcal{C}$, as a collection of clusters. For example, a possible clustering of size $k$ for a set of devices $\mathcal{D}$ might be represented as $\mathcal{C}=\{C_1, C_2, \ldots, C_k\}$. This setup partitions the entire device set $\mathcal{D}$ into distinct subsets, where the combination of individual device set $C_i.D$ collectively covers $\mathcal{D}$ without overlap. The objective of clustered federated learning is to construct a clustering $\mathcal{C}$ that minimizes the population loss $L(\mathcal{C})$ represented as follows.

$$L(\mathcal{C}) = \sum_{C \in \mathcal{C}} \sum_{d \in C.D} E_{z \sim P_d(z)}[\ell(C.\theta, z)] \tag{2}$$

This equation encapsulates the total loss across all clusters within the clustering $\mathcal{C}$, where each cluster's contribution to the loss is determined by its assigned model parameters and the data from devices within that cluster. CFL techniques generally alternate between updating the clustering and performing federated learning within each cluster to estimate the cluster model with the devices in it.

To enable asymmetric clustering, we propose to add a third component to the definition of each cluster $C_j$:

- Component 3: A set of supportive clusters, $C_j.sup$.

Supportive clusters are designed to establish inter-cluster relationships. Suppose that cluster $C_A$ requires help from another cluster, $C_B$, to train its model, then $C_B$ becomes a supportive cluster to $C_A$, indicated by $C_B \in C_A.sup$. Both members of $C_j.D$ and members of supportive clusters in $C_j.sup$ contribute to training the model $C_j.\theta$, though only the members of $C_j.D$ will use model $C_j.\theta$ for prediction. For instance, Cluster A may benefit from being grouped with Cluster B, while Cluster B prefers to remain separate. Forcing a merger between Clusters A and B could reduce the accuracy of Cluster B's model. Conversely, keeping both clusters separate does not fully utilize the available data. To address this challenge, supportive clusters are introduced. By designating Cluster B as a supportive cluster to Cluster A, members of Cluster B can assist in training Cluster A's model without directly using it for their own predictions. This approach enhances data utilization and model accuracy through inter-cluster collaboration.

## 3 RELATED WORK

Clustered Federated Learning (CFL) represents a significant advancement in managing distributed data across various devices. This subsection reviews key methodologies and their respective contributions to the field.

- **Iterative Federated Clustering Algorithm (IFCA) (Ghosh et al., 2020)** starts with a predefined number of cluster models at the server. Devices determine their cluster identity based on which models minimize their local loss.
- **Federated Stochastic Expectation Maximization (FeSEM) (Xie et al., 2021)** begins with a fixed number of clusters and iteratively assigns devices to the nearest cluster based on the $L2$ distance of the model parameters. Each cluster updates its model by averaging the models of the assigned devices.
- **FedGroup (Duan et al., 2021)** clusters devices according to their gradient cosine similarity and facilitates both inter-cluster and intra-cluster training alongside device migration.
- **FedSoft (Ruan & Joe-Wong, 2022)** operates similarly to IFCA but introduces flexibility by allowing devices to belong to multiple clusters. Each cluster's importance is determined based on the local loss for each data point.

- **FedDrift (Jothimurugesan et al., 2023)** is designed for continual learning. It starts with an assumption of homogeneous data distribution but can adapt to changes by initiating new clusters when significant shifts in data distribution are detected through loss comparison.
- **CFL-GP (Kim et al., 2024)** partitions devices into groups with similar accumulated gradients by spectral clustering.
- **SR-FCA (Vardhan et al., 2024)** successively refines a clustering of devices by bottom up aggregation based on a cross-model loss.

Despite these advancements, two primary limitations persist in CFL:

- **Fixed Number of Clusters**: Except for FedDrift and SR-FCA, most of the previous approaches use a fixed number of clusters for device grouping. This fixed cluster count presents a challenge, as it requires certain prior knowledge and needs to be accurate. If the initial guess for the number of clusters is too low, devices with varying data distributions may be incorrectly grouped together, leading to suboptimal predictors for those devices. Conversely, an excessive number of clusters can scatter devices with similar data distributions across different clusters, resulting in suboptimal predictors due to a reduced amount of data for cluster model training.
- **Symmetric Clustering Limitations**: There is no natural extension from symmetric clustering to asymmetric clustering. Devices either support each other or they do not. Traditional CFL approaches typically restrict each device to a single cluster (e.g., IFCA, FeSEM), or, as seen in soft clustering methods like FedSoft, allow devices to influence multiple clusters without adequately considering the overall impact on cluster integrity. In environments where data quality varies considerably, such strategies may compromise the robustness of clusters initially dominated by high-quality data, thus failing to balance individual benefits with collective goals effectively.

## 4 METHOD

We propose a new method called Federated Adaptive Asymmetric Clustering (FAACL) that addresses the two limitations identified in the previous section. We first describe how to initialize clusters (Sec. 4.1), merge clusters (Sec. 4.2) and train the model of each cluster (Sec. 4.3). These procedures are then combined into "flaat" and "hierarchical" versions of the FAACL algorithm (Sec. 4.4). Finally, we provide a theoretical analysis (Sec. 4.5) and discuss an approach to enhance privacy (Sec. 4.6).

### 4.1 CLUSTERING INITIALIZATION

Our approach to clustering initialization in Clustered Federated Learning adopts an intuitive strategy. In the initial phase, we create a unique cluster for each device, denoted as $C_i.D=\{d_i\}$, effectively forming singleton clusters (i.e., clusters containing only a single device). This is illustrated Algorithm 2 in Appendix A. Upon completion of the initialization, we obtain a total of $n$ such singleton clusters. Given the isolated nature of these initial clusters, the subsequent phase of our methodology focuses on the inter-device communication. This is achieved by merging similar clusters to reduce the total number of clusters, thereby enhancing the collaborative learning process among the devices.

### 4.2 CLUSTER MERGE

Following the initialization phase, which results in $n$ distinct clusters, we employ an iterative process to merge similar clusters based on support evaluations. The merging process and support evaluations are outlined in Algorithms 3 to 5 in Appendix A. Each cluster's supportive connections are updated by exchanging models among devices from different clusters.

**Model Evaluation and Support Determination**: For each pair of clusters, we assess if cluster $C_2$ supports cluster $C_1$ based on their model performances across all devices in cluster $C_1$. This is determined by comparing the model parameters $C_2.\theta$ and $C_1.\theta$. Specifically, we calculate losses for each data point $z$ in the validation set $Z_d^{val}$ for a given device $d$: $\ell(\theta_{C_1}, z_d^{val})$ and $\ell(\theta_{C_2}, z_d^{val})$, where $\ell(\theta_C, z)$ denotes the loss of model parameters $\theta_C$ on data point $z$. To assess whether cluster $C_2$ is supportive of cluster $C_1$, we implement two alternative approaches: direct mean comparison and the application of a statistical test.

**Version 1: Statistical Test Approach**: The Wilcoxon signed-rank test is employed as a non-parametric method to compare the loss distributions of $C_1.\theta$ and $C_2.\theta$. Shown in Algorithm 3, this

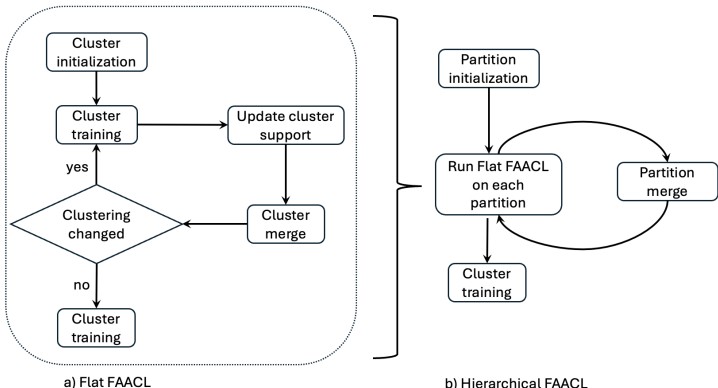

Figure 1: Flowchart for Flat FAACL (a) and hierarchical FAAL (b).

test does not require the assumption of normal distribution and is suited for paired samples:

$$H_0 : \ell(C_1.\theta, z) + \epsilon < \ell(C_2.\theta, z)$$

where $\epsilon$ is the margin of error acceptable within the hypothesis testing framework, and $p$-values smaller than a preset significance level $\alpha$ indicate significant differences between the clusters. While this method provides a practical approach to evaluate differences in model performance, it lacks the theoretical guarantees that Hoeffding's inequality provides to the direct comparison method.

**Version 2: Direct Comparison Approach**: The direct comparison approach involves calculating the average losses for model parameters $C_1.\theta$ and $C_2.\theta$ across the validation dataset for each device $d$: $\ell(\theta_{C_1}, z_d^{val})$ and $\ell(\theta_{C_2}, z_d^{val})$. If the mean difference in losses does not exceed a predetermined threshold $\epsilon$, cluster $C_2$ is considered supportive of $C_1$, illustrated in Algorithm 4. This method benefits from a theoretical guarantee provided by Hoeffding's inequality, which bounds the estimation error based on the amount of data. In Section 4.5, the application of Hoeffding's bound ensures that the direct comparison yields statistically significant results under specific conditions, providing a strong theoretical foundation for this approach.

**Cluster Merging**: Clusters that are mutually supportive are merged into a new cluster. As shown in Algorithm 5, this new cluster is formed by taking the union of the devices of the original clusters, taking the intersection of the supportive members of the original clusters and setting the parameters of the new cluster model to the parameters of any of the original cluster models.

### 4.3 CLUSTER TRAINING

During the training phase, each cluster $C_j$ engages in a series of training iterations to refine and optimize its model parameters. This optimization process is aimed at minimizing the collective loss calculated from all the data available from devices that are part of the cluster $C_j.D$ as well as data from devices belonging to supportive clusters $C_j.sup$. Any federated learning technique can be used as a subroutine to train a cluster model with its member devices and supportive devices. Algorithm 6 illustrates how to do this with the FedAvg algorithm (McMahan et al., 2017).

### 4.4 PROPOSED FEDERATED CLUSTERING METHOD

In this section, we introduce two advanced approaches for federated clustering under the FAACL framework: Flat FAACL and Hierarchical FAACL. These strategies are designed to handle the clustering of devices effectively while balancing computational efficiency and clustering performance. The runtime complexity is analyzed in Appendix A.3.

**Flat FAACL (Figure 1a)**: Flat FAACL integrates the previous algorithms into a cohesive approach. The term "flat" in this context indicates that the clustering approach treats all devices on the same level. This flat clustering processes all devices simultaneously, directly compares and merges clusters. It starts with cluster initialization, and repeatedly trains each cluster, identify support relationships and attempts to merge mutually supportive clusters. When the clustering stabilizes, further training

continues with the fixed cluster allocation, shown in Algorithm 7. This approach involves extensive pairwise interactions between clusters, leading to a computational complexity of $O(n^2)$ per iteration, where $n$ denotes the number of devices.

---

**Algorithm 1** Hierarchical FAACL

---

**Input**: Device set $\mathcal{D}$, significance level $\alpha$, threshold $\epsilon$, number of epochs $epochs$
**Output**: Clustering $\mathcal{C}$

---

Initialize partition set $\mathcal{P}=\{\}$
**for** each device $d_i \in \mathcal{D}$ **do**
    Initialize partition $P$ with $P.D=\{d_i\}$ and $P.\mathcal{C}=$None
    $\mathcal{P}.add(P)$
**while** $|\mathcal{P}| > 1$ **do**
    **for** each partition $P \in \mathcal{P}$ **do**
        $P.\mathcal{C}\leftarrow$Flat FAACL$(P.D, \alpha, \epsilon, P.\mathcal{C}, 0)$
    Initialize a new set of partitions $\mathcal{P}^*=\{\}$
    **for** $P_i, P_j$ sampled non-repeatedly from $\mathcal{P}$ **do**
        Create new partition $P'$ with $P'.D=P_i.D \cup P_j.D$ and $P'.\mathcal{C}=P_1.\mathcal{C} \cup P_2.\mathcal{C}$
        $\mathcal{P}^*.add(P')$
    $\mathcal{P}\leftarrow\mathcal{P}^*$
Let $P$ be the only remaining partition in $\mathcal{P}$, $\mathcal{C}\leftarrow P.\mathcal{C}$
**while** $epochs > 0$ **do**
    Train clusters $\mathcal{C}\leftarrow$Cluster Train$(\mathcal{C})$
    $epochs\leftarrow epochs - 1$
**Return** Final clustering $\mathcal{C}$

---

**Hierarchical FAACL (Figure 1b)**: To optimize the computational demands by Flat FAACL, we propose the Hierarchical FAACL method. This approach introduces a tiered clustering strategy, where devices are initially grouped into smaller clusters that are progressively merged to form larger clusters. This hierarchical structure significantly reduces the computational overhead by limiting the number of direct comparisons and mergers required at each stage of the process. Each level of the hierarchy forms an intermediate clustering that refines the grouping of devices, enhancing the efficiency and potentially improving the adaptability of the model to changes in device data distributions.

**Partitioned Strategy for Enhanced Efficiency (Hierarchical FAACL)**: To further enhance the efficiency of the clustering process, we employ a strategic partitioning approach, where each partition $P$ consists of a subset of devices and their associated clusterings.

- **Set of devices ($\mathbf{P.D}$)**: This subset may include devices like $\{d_1, d_2, d_3\}$, indicating the devices included in the partition.
- **Clustering formed by its device set ($\mathbf{P.\mathcal{C}}$)**: Each partition also has its own clustering, such as $\{C_1, C_2\}$, where $C_1.D = \{d_1\}$ and $C_2.D = \{d_2, d_3\}$.

The partition merging process is systematically detailed in Algorithm 1, starting with $n$ initial partitions, each containing a single device. During each iteration, two partitions, $P_i$ and $P_j$ are selected and merged to form a new partition $P'$. This new partition combines the device sets and clustering from $P_i$ and $P_j$. The combined clustering $P'.\mathcal{C}$ is then used as initial parameters $\mathcal{C}_{init}$ for the subsequent application of Flat FAACL. This iterative merging continues until only one comprehensive partition remains, effectively simplifying the clustering process while aiming to retain the efficacy of the ultimate clustering outcome. See Section A.3 for a comparison of the computational complexity of Hierarchical and Flat FAACL.

In practice, devices may enter and leave the federation at any time. When a new device appears, it is simply initialized as a singleton cluster whose model is the local model of that device. Then this new cluster participates in subsequent iterations of cluster merging and cluster updating as usual. When a device leaves the federation, it is simply removed from the support and membership of each cluster it used to contribute to. If this device was part of a singleton cluster, that cluster is deleted. Subsequent iterations of cluster merging and cluster updating proceed as usual.

### 4.5 Theoretical Guarantee

As previously discussed in the cluster support section, the Direct Comparison Approach can utilize Hoeffding's inequality to lower bound the probability of correctly clustering devices with the same data distribution.

**Theorem 4.1.** *Consider a universe of $M$ devices such that $m$ devices $d_1, ..., d_m$ each receive $n$ data points from the same underlying distribution $\mathcal{D}$ (i.e., $\forall i \in [m], Z_{d_i} \sim \mathcal{D}$). Let $error(\theta(D_1), D_2)$ denote the error of the model trained on distribution $D_1$ and inferenced on $D_2$, where error is defined as $1 - accuracy$. Let $a$ and $b$ be lower and upper bounds respectively on the difference between the predicted and actual label of each data point. Let also all other $M - m$ devices receive data from other distributions $D'$ with separation $error(\theta(\mathcal{D}'), \mathcal{D}) - error(\theta(\mathcal{D}), \mathcal{D}) \geq gap$. When flat FAACL uses $\epsilon$ as the threshold to determine support relations, then flat FAACL will cluster $d_1, ..., d_m$ together (i.e., $\exists C$ such that $\forall i \in [m], d_i \in C.D$) with probability at least $1 - \delta$.*

$$Pr(\exists C \text{ such that } \forall i \in [m], d_i \in C.D \mid \forall i \in [m], Z_{d_i} \sim \mathcal{D}) \geq 1 - \delta \qquad (3)$$

$$\text{where } \delta \leq m\left[2exp\left(-\frac{2(n\epsilon)^2}{n(b-a)^2}\right) + (M-m)exp\left(-\frac{2(n(gap-\epsilon))^2}{n(b-a)^2}\right)\right] \qquad (4)$$

A proof of this theorem is provided in the appendix A.2.

### 4.6 Privacy

Although data does not leave each device in federated learning, the models that are shared may leak information about the data used to train them (Mothukuri et al., 2021; Boenisch et al., 2023). Hence it is common to combine federated learning with a differential privacy (DP) mechanism (Wei et al., 2020) and secure multi-party computation (MPC) (Bonawitz et al., 2017; Li et al., 2020). MPC is typically used to secure the aggregation step at the server, but the resulting aggregated model may still leak data information, especially if it memorizes some of the data. Hence, to provably ensure privacy, DP is often the preferred mechanism. FAACL can be combined with a local DP mechanism (Shokri & Shmatikov, 2015) by adding noise to the weights of each model before sharing. In Algorithm 6, each device can add Gaussian noise as a function of sensitivity to achieve a desired degree of privacy at the end of each round of training. Models can then be shared with clusters and other devices while statistically preventing membership attacks at the cost of a reduction in accuracy.

## 5 Experiments

In this section, we present empirical results that validate our proposed methods within two distinct experimental scenarios: asymmetric and symmetric.

- **Asymmetric Scenario**: This scenario addresses the complexity arising from the diversity in device data distributions. It explores situations where devices are not reciprocal.
- **Symmetric Scenario**: Devices either share identical data distributions or possess completely contrasting distributions. The formation of clusters is such that devices within the same cluster either mutually benefit or detrimentally affect each other's model training outcomes. Optimal performance is achieved when devices with similar distributions are clustered together, and those with divergent distributions are separated.

This section is structured as follows: We start by describing the baselines and datasets in Section 5.1, explore asymmetric and symmetric settings in Section 5.2, 5.3. Finally, we synthesize our findings and discuss their implications in Section 5.4.

### 5.1 Baselines and Datasets

In our empirical evaluation, we compare our algorithms against well-established methods in Federated Learning: Centralized FL, FedAvg (McMahan et al., 2017), FedGroup(Duan et al., 2021), IFCA (Ghosh et al., 2020), FeSEM (Xie et al., 2021), FedDrift (Jothimurugesan et al., 2023), CFL-GP (Kim et al., 2024) and SR-FCA (Vardhan et al., 2024).

Our experiments span several datasets known for their applicability in federated learning research: MNIST (Deng, 2012), Extended MNIST (EMNIST) (Cohen et al., 2017), Fashion MNIST (FASH-ION) (Xiao et al., 2017), Federated Extended MNIST (FEMNIST) (Caldas et al., 2018), CIFAR10 (Krizhevsky et al., 2009) and Sentiment140 (SENT140) (Caldas et al., 2018). These datasets provide a diverse set of challenges and allow for a rigorous assessment of our algorithm's performance across both image and text classification tasks.

In our experiments, we implemented the hierarchical version of our FAACL algorithm. This choice was made because, while the hierarchical and flat versions theoretically converge to the same clustering outcomes (in the limit of infinite data), the hierarchical approach is more computationally efficient as shown in Table 10 of Appendix C.7. The details of the baseline methodologies, the datasets employed, and the hyperparameters used in our experiments, are provided in Appendix C. This supplementary section contains detailed information that supports the reproducibility of our experimental procedures. We propose two versions of the FAACL method for cluster support:

**Version 1** employs the Wilcoxon signed-rank test to evaluate support between clusters, making it suitable for data with non-normal loss distributions. It leverages statistical testing to ensure that differences in model performance are statistically insignificant before considering clusters supportive.

**Version 2** utilizes direct comparison of mean losses between clusters, applying Hoeffding's bound to provide error estimation guarantees. It is beneficial when theoretical robustness is required, determining support based on if mean loss differences fall within a predefined acceptable range.

### 5.2 ASYMMETRIC SETTING

In the asymmetric setting, we explore a more complex scenario where devices' data distributions are not entirely contradictory, and some devices may benefit from collaborating with others from different distributions. This setting is designed to showcase the adaptability and performance advantages of our proposed federated learning technique in creating asymmetric clusters. We design experiments with both natural and synthetic partitions with respect to data quality and quantity.

**Natural Data Partition ($A_0$)** Utilizing the FEMNIST subset of the LEAF dataset, we simulate a realistic asymmetric scenario. For instance, in the FEMNIST dataset, each device corresponds to a unique writer, allowing us to simulate a realistic asymmetric scenario. Given that different writers produce images of varying quality for classification, asymmetric clustering becomes beneficial. In such a setup, devices associated with high-quality writers can lend support to those linked to lower-quality writers, demonstrating the utility of asymmetric clusters in enhancing overall performance.

**Synthetic Image Partition ($A_1$)** In this setup, one group of devices receives data with pristine quality (no noise), while another group handles data contaminated with Gaussian (Shannon, 1948) and salt & pepper noise (Castleman, 1996). This partition serves to illustrate the need for asymmetric clustering, where devices with noisy data can benefit from the cleaner inputs of other devices. It tests the algorithm's ability to optimize learning outcomes in the presence of varying data quality.

**Synthetic Data Amount Partition ($A_2$)** We construct a scenario where one set of devices has access to abundant data, while another set is limited in data quantity and exhibits slightly different predictors. Asymmetric clustering plays a crucial role in such environments. Devices with enough data, although reluctant to merge due to predictor discrepancies, can still offer valuable insights to devices with sparse data. Conversely, devices with limited data can leverage the more extensive datasets of others to reduce variance, even at the risk of introducing some bias.

### 5.3 SYMMETRIC SETTING

In the symmetric setting, our objective is to evaluate the adaptability and effectiveness of our proposed federated learning technique under conditions of either homogeneous or extremely heterogeneous data distributions among devices. We conduct these evaluations using both natural and synthetic data partitions to simulate various distribution scenarios.

**Natural Data Partition ($S_0$):** This setup simulates a scenario where each device's data is independently and identically distributed. This setup tests the algorithm's ability to recognize and maintain uniformity across devices in a federated environment.

**Synthetic Label Partition ($S_1$):** In this setup, the dataset is allocated among devices based on distinct label ranges. For example, one set of devices might exclusively receive data corresponding to labels 0-4, while another set of devices receives data with labels 5-9. In this scenario, since devices with different label distributions are less likely to benefit each other, forming separate clusters for each label distribution is considered optimal.

**Synthetic Predictor Partition ($S_2$):** In this partition, datasets of devices from different distributions have different underlying predictors. For instance, in the MNIST dataset, one group of devices (set 1) may map images directly to their corresponding labels, while another group (set 2) maps images to shifted labels (e.g., mapping the image of digit 0 to label 1). This setup challenges the algorithm's ability to handle scenarios with significant variations in data mappings across devices. In such cases, combining data from different distributions can be harmful, indicating the need for distinct clusters to maintain the integrity of each device's predictive model.

## 5.4 EXPERIMENTAL RESULTS

The outcomes of our experiments are presented across Tables 1-4, and the full tables with standard error are presented in Appendix C.7. In both asymmetric and symmetric settings, we conducted experiments under both natural and synthetic data distribution scenarios. For the baseline setups, we begin with an initial allocation of five clusters, which is chosen based on the anticipated diversity within each dataset's data distributions. We set this number as an upper bound for the potential number of clusters, ensuring that the model has the capacity to accurately represent all possible clusters. This setup mirrors conditions in real-world federated learning scenarios, where the exact number of natural data clusters is unknown. By initializing more clusters, the baseline algorithms retain the flexibility to achieve accurate clustering by potentially leaving some clusters empty.

Table 1: Test accuracies $\pm$ stderr with [number of clusters] in $A_0$.

| Dataset | FEMNIST | SENT140 |
|---|---|---|
| IFCA | $51.21\pm0.38$ [5] | $72.86\pm0.24$ [5] |
| FeSEM | $47.52\pm3.92$ [1] | $63.69\pm1.83$ [2] |
| FedGroup | $65.47\pm1.02$ [5] | $72.38\pm0.43$ [5] |
| FedDrift | $62.28\pm0.39$ [8] | $73.86\pm0.68$ [14] |
| FedSoft | $67.87\pm0.91$ [5] | $72.26\pm1.12$ [5] |
| CFL-GP | $66.38\pm0.73$ [5] | $73.21\pm0.48$ [5] |
| SR-CFA | $68.32\pm0.28$ [7] | $74.12\pm0.62$ [9] |
| FAACL(version 1) | $\mathbf{71.34\pm0.07}$ [15] | $76.24\pm0.83$ [10] |
| FAACL(version 2) | $70.35\pm0.32$ [7] | $\mathbf{76.38\pm0.52}$ [6] |

**Asymmetric Natural Distribution ($A_0$)**: In the natural distribution, the FEMNIST and SENT140 dataset from the LEAF benchmark are used to construct each device to represent a writer / user. In this natural distribution, the true number of clusters is unknown, we train the centralized method with all devices in one cluster. In this experiment with real data (Table 1), our methods achieved higher accuracy by at least 5% compared to other clustered baselines, including FedDrift and SR-CFA which can form adaptive clusters. This improvement in the real-world datasets underscores the advantage of asymmetric clustering.

**Asymmetric Synthetic Distribution ($A_1, A_2$)**: In the synthetic experiments (Table 2), we first report upper bounds for a centralized technique and a decentralized technique (FedAvg optimal) that is given the true underlying clusters. Since all techniques use FedAvg internally to aggregate models, but differ in how they estimate clusters, the gap in performance between each technique and FedAvg optimal corresponds to the loss in accuracy due to suboptimal clustering. Furthermore, the gap between FedAvg optimal and Centralize is the loss due to decentralize learning. Our approach consistently outperforms the other clustered FL techniques since constructing an asymmetric clustering consistently maintains a cluster for each device for prediction purposes. Each device can contribute to the training of other clusters without worrying about hurting its performance.

**Symmetric Natural Distribution ($S_0$)**: In scenarios with natural data distributions (Table 3), our method, FAACL, demonstrates performance that is comparable to other baselines. It is important to note that other clustered federated learning methods, including IFCA, FeSEM, FedSoft, FedGroup and CFL-GP are initialized with a fix number (5) of clusters, but may converge to a smaller number of clusters (number in brackets) by leaving some clusters empty. This can lead to under-trained models due to data dilution across too many clusters, especially when the final number of utilized clusters does not align with the optimal cluster count for the given data distribution. Although FedDrift is capable of dynamically determining the number of clusters during training, it still tends to output more clusters.

**Symmetric Synthetic Distribution ($S_1, S_2$)**: In the synthetic distribution (Table 4) (i.e., synthetic labels and synthetic predictors), our method still outperforms other baselines in most datasets due to its adaptive number of clusters.

A challenge in the asymmetric setting is determining the threshold between different distributions. Given the divergence in data distributions and the varying impact of clustering on different devices, finding the ideal number of clusters can be non-trivial. However, the experiments show that our method's capability to perform asymmetric clustering, while not necessarily finding the number of correct clusters, consistently delivers the best accuracy in various settings.

## 6 CONCLUSION

In this study, we introduced the novel concept of asymmetric clustering to address scenarios in federated learning where clustering benefits are unevenly distributed—some devices benefit from joining a new cluster, while others may experience detrimental effects. To tackle this challenge,

we developed the Federated Adaptive Asymmetric Clustering Learning (FAACL) method, which facilitates the formation of asymmetric clusters.

We adopted a partition-based strategy to circumvent the complexities associated with determining adaptive clustering, effectively reducing the device complexity to $O(n^2)$ for the entire process within $O(\log n)$ iterations. Our empirical evaluations underscore FAACL's superior performance in comparison to conventional "symmetric" clustering approaches, particularly with real-world datasets. The results not only validate the effectiveness of FAACL, but also spotlight the promising potential of asymmetric clustering in practical federated learning applications, setting the stage for future advancements in the field.

Table 2: Test accuracies for $A_1, A_2$ with [number of clusters]. Centralize and FedAvg (optimal) are upper bounds that use the true underlying clusters. Stderr is reported in Tables 11, 12.

| Dataset | MNIST | | EMNIST | | FASHION | | CIFAR10 | |
|---|---|---|---|---|---|---|---|---|
| | $A_1$ | $A_2$ | $A_1$ | $A_2$ | $A_1$ | $A_2$ | $A_1$ | $A_2$ |
| Centralize | 78.39[2] | 97.32[2] | 62.39[2] | 97.67[2] | 73.43[2] | 86.90[2] | 58.51[2] | 64.16[2] |
| FedAvg (optimal) | 77.58[2] | 97.10[2] | 60.25[2] | 97.38[2] | 73.32[2] | 86.72[2] | 57.81[2] | 64.04[2] |
| IFCA | 69.90[5] | 95.35[3] | 50.50[5] | 95.98[2] | 69.09[4] | 85.70[4] | 53.29[5] | 59.17[5] |
| FeSEM | 65.79[4] | 74.65[1] | 46.31[1] | 88.02[1] | 64.56[1] | 81.27[1] | 48.72[1] | 52.84[5] |
| FedGroup | 74.65[5] | 95.44[5] | 51.88[5] | 95.42[5] | 70.93[5] | 86.31[5] | 54.26[5] | 60.27[5] |
| FedDrift | 70.34[7] | 95.34[3] | 51.63[4] | 91.70[4] | 72.96[3] | 84.15[2] | 54.58[1] | 60.28[8] |
| FedSoft | 71.28 [5] | 95.73 [5] | 52.38[5] | 95.73 [5] | 71.01 [5] | 84.29 [5] | 53.38[5] | 60.25[5] |
| CFL-GP | 72.16 [5] | 94.36 [5] | 52.31 [5] | 94.24 [5] | 70.39 [5] | 83.25 [5] | 52.14[5] | 61.17 [5] |
| SR-CFA | 74.27 [4] | 95.38 [2] | 53.84 [6] | 96.48 [5] | 70.04 [8] | 84.57 [5] | 51.62[4] | 61.09 [5] |
| FAACL(version 1) | **76.31**[4] | **96.40**[5] | **57.28**[9] | 96.95[4] | **73.21**[3] | **86.53**[2] | **56.31**[6] | **62.74**[8] |
| FAACL(version 2) | 75.12[4] | 96.06[4] | 55.85[3] | **97.02**[3] | 73.19[3] | 86.50[2] | 55.42[4] | 62.43[4] |

Table 3: Test accuracies $\pm$ stderr with [number of clusters] in $S_0$. Centralize and FedAvg (optimal) are upper bounds that use the true underlying clusters.

| Dataset | MNIST | EMNIST | FASHION | CIFAR10 |
|---|---|---|---|---|
| Centralize | 97.64$\pm$0.02 [1] | 98.08$\pm$0.05 [1] | 89.21$\pm$0.07 [1] | 76.42$\pm$0.06 [1] |
| FedAvg (optimal) | 96.24$\pm$0.10 [1] | 97.71$\pm$0.08 [1] | 88.63$\pm$0.21 [1] | 73.28$\pm$0.13 [1] |
| IFCA | 95.05$\pm$0.23 [3] | 94.54$\pm$0.83 [1] | 85.67$\pm$0.38 [4] | 71.46$\pm$0.37 [5] |
| FeSEM | 94.29$\pm$0.44 [3] | 94.20$\pm$1.13 [1] | 86.33$\pm$0.18 [1] | 68.43$\pm$0.16 [3] |
| FedGroup | 95.78$\pm$0.23 [5] | 96.09$\pm$0.20 [5] | 86.19$\pm$0.08 [5] | 72.27$\pm$0.14 [5] |
| FedDrift | 95.66$\pm$0.82 [2] | 97.07$\pm$0.26 [3] | 86.98$\pm$0.67 [3] | 71.47$\pm$0.50 [2] |
| FedSoft | 96.03$\pm$0.21 [5] | 93.21$\pm$0.19 [5] | 83.58$\pm$0.22 [5] | **72.74**$\pm$**0.18** [5] |
| CFL-GP | **96.28**$\pm$**0.43** [5] | 97.29$\pm$0.35 [5] | 86.39$\pm$0.63 [5] | 71.08$\pm$0.25 [5] |
| SR-CFA | 95.89$\pm$0.74 [1] | 96.36$\pm$0.52 [1] | 84.38$\pm$0.71 [1] | 70.49$\pm$0.18 [1] |
| FAACL(version 1) | 96.14$\pm$0.99 [1] | 97.22$\pm$0.28 [1] | **88.25**$\pm$**0.44** [1] | 72.57$\pm$0.23 [1] |
| FAACL(version 2) | 96.07$\pm$0.28 [1] | **97.31**$\pm$**0.72** [1] | 87.73$\pm$0.81 [1] | 72.60$\pm$0.60 [1] |

Table 4: Test accuracies for $S_1, S_2$ with [number of clusters]. Centralize and FedAvg (optimal) are upper bounds that use the true underlying clusters. Stderr is reported in Tables 13, 14.

| Dataset | MNIST | | EMNIST | | FASHION | | CIFAR10 | |
|---|---|---|---|---|---|---|---|---|
| | $S_1$ | $S_2$ | $S_1$ | $S_2$ | $S_1$ | $S_2$ | $S_1$ | $S_2$ |
| Centralize | 94.85[2] | 97.06[2] | 97.11[2] | 96.93[2] | 90.82[2] | 88.95[2] | 74.68[2] | 73.57[2] |
| FedAvg (optimal) | 94.39[2] | 96.73[2] | 97.09[2] | 96.80[2] | 90.47[2] | 88.74[2] | 74.50[2] | 73.18[2] |
| IFCA | 91.16[5] | 94.36[4] | 94.28[4] | 95.17[2] | 86.88[4] | 85.42[5] | 69.36[5] | 71.11[4] |
| FeSEM | 50.24[3] | 49.35[3] | 42.44[1] | 43.94[1] | 50.57[1] | 43.65[1] | 54.20[1] | 64.76[3] |
| FedGroup | **93.73**[5] | 95.55[5] | 96.18[5] | 95.79[5] | 88.50[5] | 85.98[5] | **71.62**[5] | 70.81[5] |
| FedDrift | 91.76[8] | 93.37[6] | 96.35[3] | 96.12[3] | 85.52[7] | 85.77[4] | 70.53[3] | 71.30[3] |
| FedSoft | 90.49 [5] | 93.92 [5] | 94.39 [5] | 94.78 [5] | 84.29 [5] | 85.11 [5] | 72.49[5] | **71.58**[5] |
| CFL-GP | 92.31 [5] | 94.92 [5] | 96.73 [5] | 95.75 [5] | 88.29 [5] | 86.02 [5] | 70.35[5] | 71.21[5] |
| SR-CFA | 92.16 [2] | 95.06 [2] | 96.01 [2] | 95.88 [2] | 89.26 [2] | 86.69 [2] | 70.23[2] | 71.47[2] |
| FAACL(version 1) | 93.45[2] | **95.89**[2] | **96.82**[2] | 96.54[2] | **90.23**[2] | **87.82**[2] | 71.48[2] | 71.47[2] |
| FAACL(version 2) | 93.27[2] | 95.71[2] | 96.44[2] | **96.62**[2] | 89.64[2] | 87.31[2] | 71.27[2] | 71.43[2] |

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

## A  ALGORITHM ANALYSIS

### A.1  ALGORITHM PSEUDOCODE

---

**Algorithm 2** Clustering Initialization

---

**Input**: Device set of size n, $\mathcal{D} = \{d_1, \ldots, d_n\}$
**Output**: Initial clustering $\mathcal{C}$

---

**Server**: Initialize $\mathcal{C} = \{\}$ and random parameters $\theta$
**Server**: Distribute $\theta$ to all devices in $\mathcal{D}$
**for** each device $d_i \in \mathcal{D}$ **do**
    **Device**: Form a new cluster $C_i$ with $C_i.D = \{d_i\}$, $C_i.sup = \{\}$
    **Device**: Initialize parameters $C_i.\theta = \theta$ and $C_i.\theta = argmin_\theta \sum_{z \in Z_{d_i}^{train}} \ell(\theta, z)$
    **Device**: Send the cluster $C_i$ to server
**Server**: $\mathcal{C} \leftarrow \{C_i\}_{i=1}^n$
**Return** Initial clustering $\mathcal{C}$

---

**Algorithm 3** Cluster Support (Version 1)

---

**Input**: Clustering $\mathcal{C}$, significance level $\alpha$, threshold $\epsilon$
**Output**: Updated Clustering $\mathcal{C}$

---

**for** each cluster $C_1 \in \mathcal{C}$ **do**
    **for** each distinct cluster $C_2 \in \mathcal{C}$ **do**
        **Server**: Initialize $support\_flag \leftarrow True$
        **for** each device $d \in C_1.D$ **do**
            **Server**: Send parameters $C_1.\theta, C_2.\theta$ to device $d$
            **Device**: $L \leftarrow \{(\ell(C_1.\theta, z), \ell(C_2.\theta, z)) | z \in Z_d^{val}\}$
            **Device**: Computes $p \leftarrow Wilcoxon(L, \epsilon)$
            **if** $p > \alpha$ **then**
                **Server**: $support\_flag \leftarrow False$ and **break**
        **if** $support\_flag$ is True **then**
            **Server**: $C_1.sup.add(C_2)$
**Return** Updated clustering $\mathcal{C}$

---

**Algorithm 4** Cluster Support (Version 2)

---

**Input**: Clustering $\mathcal{C}$, significance level $\alpha$, threshold $\epsilon$
**Output**: Updated Clustering $\mathcal{C}$

---

**for** each cluster $C_1 \in \mathcal{C}$ **do**
    **for** each distinct cluster $C_2 \in \mathcal{C}$ **do**
        **Server**: Initialize $support\_flag \leftarrow True$
        **for** each device $d \in C_1.D$ **do**
            **Server**: Send parameters $C_1.\theta, C_2.\theta$ to device $d$
            **Device**: $L_d^1 \leftarrow \{\ell(C_1.\theta, z) | z \in Z_d^{val}\}$, $L_d^2 \leftarrow \{\ell(C_2.\theta, z) | z \in Z_d^{val}\}$ uploaded to server
        **Server**: Combine $L \leftarrow \{L_1, L_2\}$, where $L_1 = \{L_d^1 : d \in C_1.D\}$, $L_2 = \{L_d^2 : d \in C_1.D\}$
        **Server**: Compute $p = Wilcoxon(L)$
        **if** $p > \alpha$ **then**
            **Server**: $support\_flag \leftarrow False$ and **continue**
        **if** $support\_flag$ is **True then**
            **Server**: $C_1.sup.add(C_2)$
**Return** Updated clustering $\mathcal{C}$

---

---

**Algorithm 5** Cluster Merge

---

**Input**: Clustering $\mathcal{C}$
**Output**: Refined Clustering $\mathcal{C}$

---

**Server**: Let $\mathcal{C}_{new}$ and $\mathcal{C}_{old}$ be empty
**for** each cluster $C_1 \in \mathcal{C}$ not in $\mathcal{C}_{old}$ **do**
    **for** each distinct cluster $C_2 \in \mathcal{C}$ not in $\mathcal{C}_{old}$ **do**
        **if** $C_1 \in C_2.sup$ and $C_2 \in C_1.sup$ **then**
            **Server**: Form new cluster $C'$
            $C'.D = C_1.D \cup C_2.D$, $C'.sup = C_1.sup \cap C_2.sup$, $C'.\theta = random(C_1.\theta, C_2.\theta)$
            **Server**: $\mathcal{C}_{new}.add(C'), \mathcal{C}_{old}.add(C_1, C_2)$
**Server**: $\mathcal{C} \leftarrow \mathcal{C} - \mathcal{C}_{old} + \mathcal{C}_{new}$
**Return** Refined clustering $\mathcal{C}$

---

**Algorithm 6** Cluster Train

---

**Input**: Clustering $\mathcal{C}$
**Output**: Updated Clustering $\mathcal{C}$

---

**for** each cluster $C \in \mathcal{C}$ **do**
    **for** each device $d \in C.D$ **do**
        **Server**: Send cluster model parameters $C.\theta$ to $d$
        **Device**: $C_d.\theta \leftarrow C.\theta - \lambda \nabla(\sum_{z \in Z_d^{train}} loss(C.\theta, z))$ (gradient descent)
        **Device**: Send $C_d.\theta$ and $|Z_d^{train}|$ to server
    **for** each supportive cluster $C' \in C.sup$ **do**
        **for** each device $d' \in C'.D$ **do**
            **Server**: Send cluster model parameters $C.\theta$ to $d'$
            **Device**: $C_{d'}.\theta \leftarrow C.\theta - \lambda \nabla(\sum_{z \in Z_{d'}^{train}} loss(C.\theta, z))$ (gradient descent)
            **Device**: Send $C_{d'}.\theta$ and $|Z_{d'}^{train}|$ to server
    **Server**: $C.\theta \leftarrow \frac{\sum_{d \in C.D}(|Z_d^{train}| \times C_d.\theta) + \sum_{C' \in C.sup} \sum_{d' \in C'.D}(|Z_{d'}^{train}| \times C_{d'}.\theta)}{\sum_d |Z_d^{train}| + \sum_{C' \in C.sup} \sum_{d' \in C'.D} |Z_{d'}^{train}|}$ (weighted average)
**Return** Updated clustering $\mathcal{C}$

---

**Algorithm 7** Flat FAACL

---

**Input**: Device set $\mathcal{D}$, significance level $\alpha$, threshold $\epsilon$, initial clustering $\mathcal{C}_{init}$, number of epochs $epochs$
**Output**: Optimized Clustering $\mathcal{C}$

---

Initialize $\mathcal{C} = \{\}$
**if** $\mathcal{C}_{init}$ are given **then**
    $\mathcal{C}^* = \mathcal{C}_{init}$
**else**
    $\mathcal{C}^* =$ Cluster Initialization$(\mathcal{D})$
**while** $\mathcal{C} \neq \mathcal{C}^*$ **do**
    Update $\mathcal{C} \leftarrow \mathcal{C}^*$
    Train clusters $\mathcal{C}^* \leftarrow$ Cluster Train$(\mathcal{C}^*)$
    Update cluster support $\mathcal{C}^* \leftarrow$ Cluster Support$(\mathcal{C}^*, \alpha, \epsilon)$
    Merge clusters $\mathcal{C}^* \leftarrow$ Cluster Merge$(\mathcal{C}^*)$
**while** $epochs > 0$ **do**
    Train clusters $\mathcal{C} \leftarrow$ Cluster Train$(\mathcal{C})$
    $epochs \leftarrow epochs - 1$
**Return** Final clustering $\mathcal{C}$

---

## A.2 PROOF OF THEOREM 4.1

For convenience we repeat the statement of Theorem 4.1 and then describe its proof.

**Theorem 4.1** Consider a universe of $M$ devices such that $m$ devices $d_1, ..., d_m$ each receive $n$ data points from the same underlying distribution $\mathcal{D}$ (i.e., $\forall i \in [m], Z_{d_i} \sim \mathcal{D}$). Let $error(\theta(D_1), D_2)$ denote the error of the model trained on distribution $D_1$ and inferenced on $D_2$, where error is defined as $1 - accuracy$. Let $a$ and $b$ be lower and upper bounds respectively on the difference between the predicted and actual label of each data point. Let also all other $M - m$ devices receive data from other distributions $D'$ with separation $error(\theta(\mathcal{D}'), \mathcal{D}) - error(\theta(\mathcal{D}), \mathcal{D}) \geq gap$. When flat FAACL uses $\epsilon$ as the threshold to determine support relations, then flat FAACL will cluster $d_1, ..., d_m$ together (i.e., $\exists C$ such that $\forall i \in [m], d_i \in C.D$) with probability at least $1 - \delta$.

$$Pr(\exists C \text{ such that } \forall i \in [m], d_i \in C.D \mid \forall i \in [m], Z_{d_i} \sim \mathcal{D}) \geq 1 - \delta \tag{5}$$

$$\text{where } \delta \leq m \left[ 2exp\left( -\frac{2(n\epsilon)^2}{n(b-a)^2} \right) + (M - m)exp\left( -\frac{2(n(gap - \epsilon))^2}{n(b-a)^2} \right) \right] \tag{6}$$

*Proof.* Let $X_1, \ldots, X_n$ denote the random variables that represent the error difference between model $A$ and model $B$ on each datapoint of $B$. When the error difference exceeds $\epsilon$ (i.e., $X_i > \epsilon$) then $A$ does not support $B$, otherwise, denote $A$ support $B$

Consider Hoeffding's inequality:

$$P(S_n - E[S_n] > t) \leq exp\left( -\frac{2t^2}{n(b-a)^2} \right) \tag{7}$$

$$P(E[S_n] - S_n > t) \leq exp\left( -\frac{2t^2}{n(b-a)^2} \right) \tag{8}$$

where $S_n$ is the sum of $n$ i.i.d. random variables, each lower bounded by $a$ and upper bounded by $b$. We can use Hoeffding's inequality to bound the probability with which we will make an error regarding each support relation. Consider two models $A$ and $B$ for which we would like to test whether $A$ supports $B$.

Suppose that $A$ supports $B$. This means that $E[S_n]/n \leq \epsilon$. The probability that flat FAACL makes a mistake (i.e., $S_n/n \geq \epsilon$) can be computed using Hoeffding's bound as follows:

$$P(\frac{S_n}{n} = \epsilon + t | \text{A supports B}) \tag{9}$$

$$= P(\frac{S_n}{n} = \epsilon + t | \frac{E[S_n]}{n} \leq \epsilon) \tag{10}$$

$$= P(\frac{S_n}{n} - \frac{E[S_n]}{n} \geq t | \frac{E[S_n]}{n} \leq \epsilon) \tag{11}$$

$$\leq exp\left( -\frac{2(nt)^2}{n(b-a)^2} \right) \tag{12}$$

Suppose that $A$ does not support $B$. This means that $E[S_n]/n \geq \epsilon$. The probability that FAACL makes a mistake (i.e., $S_n/n \leq \epsilon$) can be computed using Hoeffding's bound as follows:

$$P(\frac{S_n}{n} = \epsilon - t | \text{A does not supports B}) \tag{13}$$

$$= P(\frac{S_n}{n} = \epsilon - t | \frac{E[S_n]}{n} \geq \epsilon) \tag{14}$$

$$= P(\frac{E[S_n]}{n} - \frac{S_n}{n} \geq t | \frac{E[S_n]}{n} \geq \epsilon) \tag{15}$$

$$\leq exp\left( -\frac{2(nt)^2}{n(b-a)^2} \right) \tag{16}$$

Note that Flat FAACL constructs clusters in stages. Initially, each device is its own cluster. Then at each stage, Flat FAACL checks support relations between every pair of clusters and merges a pair of clusters when the two clusters support each other. Suppose that $d_i, d_j \in C$ are compared. The probability that Flat FAACL makes a mistake can be bounded as follows:

$$P(d_i, d_j \text{ are not clustered } | d_i, d_j \in C) \tag{17}$$

$$= P(d_i \text{ does not support } d_j \text{ or } d_j \text{ does not support } d_i | d_i, d_j \in C) \tag{18}$$

$$\leq P(d_i \text{ does not support } d_j | d_i, d_j \in C) + P(d_j \text{ does not support } d_i | d_i, d_j \in C) \tag{19}$$

$$\leq 2exp\left(-\frac{2(n\epsilon)^2}{n(b-a)^2}\right) \tag{20}$$

Similarly, consider $d_i \in C$ and $d_j \notin C$. Then the probability that Flat FAACL makes a mistake by clustering them together is:

$$P(d_i, d_j \text{ are clustered } | d_i \in C, d_j \notin C) \tag{21}$$

$$= P(d_i \text{ supports } d_j \text{ and } d_j \text{ supports } d_i | d_i \in C, d_j \notin C) \tag{22}$$

$$\leq \max\{P(d_i \text{ supports } d_j | d_i \in C, d_j \notin C), P(d_j \text{ supports } d_i | d_i \in C, d_j \notin C)\} \tag{23}$$

$$\leq exp\left(-\frac{2(n(gap-\epsilon))^2}{n(b-a)^2}\right) \tag{24}$$

At every iteration, Flat FAACL may mistakenly cluster $d_i \in C$ with some $d_j \notin C$ with the following probability

$$P(\exists j \text{ such that } d_i, d_j \text{ are clustered } | d_i \in C, d_j \notin C) \tag{25}$$

$$= P(\exists j \text{ such that } d_i \text{ supports } d_j \text{ and } d_j \text{ supports } d_i | d_i \in C, d_j \notin C) \tag{26}$$

$$\leq \sum_{j \in \{m+1, M\}} P(d_i \text{ supports } d_j \text{ and } d_j \text{ supports } d_i | d_i \in C, d_j \notin C) \tag{27}$$

$$\leq (M - m)exp\left(-\frac{2(n(gap-\epsilon))^2}{n(b-a)^2}\right) \tag{28}$$

At every iteration, Flat FAACL may miss the opportunity to cluster $d_i \in C$ with some $d_j \in C$ with the following probability:

$$P(\forall j \neq i \text{ such that } d_i, d_j \text{ are not clustered } | d_i, d_j \in C) \tag{29}$$

$$\leq \max_{j \neq i} P(d_i, d_j \text{ are not clustered } | d_i, d_j \in C) \tag{30}$$

$$= P(d_i, d_j \text{ are not clustered } | d_i, d_j \in C) \tag{31}$$

$$\leq 2exp\left(-\frac{2(n\epsilon)^2}{n(b-a)^2}\right) \tag{32}$$

Overall, the probability that Flat FAACL will make a mistake with respect to $d_i \in C$ by not clustering it with any other $d_j \in C$ or clustering it with any $d_j \notin C$ is:

$$P(\text{mistake regarding } d_i | d_i \in C) \leq 2exp\left(-\frac{2(n\epsilon)^2}{n(b-a)^2}\right) + (M - m)exp\left(-\frac{2(n(gap-\epsilon))^2}{n(b-a)^2}\right) \tag{33}$$

Since it will take $\log m$ iterations to form $C$ and the number of subclusters of $C$ for which Flat FAACL may make a mistake at each iteration is $m/2^i$ then the overall probability $\delta$ of making a mistake is:

$$\delta \leq \left(\sum_{i=1}^{\log m} \frac{m}{2^i}\right)\left[2exp\left(-\frac{2(n\epsilon)^2}{n(b-a)^2}\right) + (M - m)exp\left(-\frac{2(n(gap-\epsilon))^2}{n(b-a)^2}\right)\right] \tag{34}$$

$$\leq m\left[2exp\left(-\frac{2(n\epsilon)^2}{n(b-a)^2}\right) + (M - m)exp\left(-\frac{2(n(gap-\epsilon))^2}{n(b-a)^2}\right)\right] \tag{35}$$

$\square$

## A.3 COMPLEXITY ANALYSIS

The complexity of each clustering algorithm within our framework is analyzed to understand the computational demands of the process.

**Proposition A.1.** *The Cluster Initialization algorithm operates with a complexity of $O(n)$.*

*Proof.* Creating one cluster per device for $n$ devices directly results in a complexity of $O(n)$, as detailed in Algorithm 2. $\square$

**Proposition A.2.** *The Cluster Support algorithm operates with a complexity of $O(n^2)$.*

*Proof.* With $m=O(n)$ clusters, each cluster's potential supportive clusters are assessed among all others, resulting in a complexity bounded by the product of the number of clusters $m=O(n)$ and the maximum number of devices per cluster $O(n)$, yielding $O(n^2)$. $\square$

**Proposition A.3.** *The Cluster Merge algorithm operates with a complexity of $O(n^2)$.*

*Proof.* With $m=O(n)$ clusters, ths algorithm goes through each pair of cluster, resulting in a complexity $O(m^2)=O(n^2)$. $\square$

**Proposition A.4.** *The Cluster Training algorithm operates with a complexity of $O(n^2)$.*

*Proof.* Given $m=O(n)$ clusters, the training process for a cluster model involves all devices in the cluster, along with all devices from its supportive clusters. For a cluster $C$, the number of devices participating its training process is at most $|C.D| + \sum_{C' \in C.sup} |C'.D|$, thus the overall complexity is $\sum_{i=1}^{m}(|C_i.D| + \sum_{C' \in C_i.sup} |C'.D|) = \sum_{i=1}^{m} |C_i.D| + \sum_{i=1}^{m} \sum_{C' \in C_i.sup} |C'.D| = n + mn = O(n^2)$. $\square$

**Proposition A.5.** *The Flat FAACL algorithm operates with a per-iteration complexity of $O(n^2)$. When provided with an initial clustering of size $O(1)$, the total computational complexity remains $O(n^2)$. When not provided with initial clustering, the total computational complexity is $O(n^3)$.*

*Proof.* Starting with an initialization phase of $O(n)$ complexity, Flat FAACL involves training and merging phases within each iteration, both of which contribute to a per-iteration complexity of $O(n^2)$ from the Proposition A.1, A.2, A.3, A.4. Given that the algorithm's convergence criteria are met within a finite number of iterations, and assuming the initial clustering involves a minimal number of clusters ($O(1)$), Flat FAACL effectively operates with an overall complexity of $O(n^2)$. This is due to the fact that the number of iterations required for convergence does not significantly alter the computational load, which is dominated by the costs of training and merging operations within each iteration. When the initial clustering is not provided, as the initial clustering has size of $O(n)$, the total iteration is $O(n)$, therefore the total complexity is $O(n^3)$. $\square$

**Theorem A.6.** *Hierarchical FAACL reduces the overall complexity to $O(n^2)$, in $\log n$ iterations of $O(n^2 / \log n)$ complexity each.*

*Proof.* Starting from $n$ initial partitions, the algorithm progressively merges these partitions, halving their number each iteration, requiring a total of $\log n$ iterations. At the iteration $i$, there are $\frac{n}{2^i}$ partitions, each potentially containing up to $2^i$ devices. A merged partition $P'$ would have set of devices of size $2^{i+1}$, and initial clustering of size $O(1)$. By Proposition A.5, the overall complexity of applying Flat FAACL to a new partition is $O(2^i)^2$. As there are total of $\frac{n}{2^i}$ partitions, the per-iteration complexity is $O(2^i)^2 \times \frac{n}{2^i} = O(2^i \cdot n)$. Then the overall complexity until convergence is $\sum_{i=1}^{\log n} O(2^i \cdot n) = O(n^2)$. $\square$

In summary, the complexity analysis underscores Hierarchical FAACL's efficiency at managing computational resources and adapting to the scale of federated learning environments. In contrast to Flat FAACL, which faces a potential complexity of up to $O(n^3)$ due to its $O(n)$ iterations, Hierarchical FAACL reduces this complexity to $O(n^2)$, ensuring completion within just $\log n$ iterations. By reducing the complexity and leveraging a logarithmic number of iterations, Hierarchical

FAACL offers a scalable and efficient solution to clustering in large, distributed networks. In comparison, all clustered federated learning techniques have a complexity of least $O(n|\mathcal{C}|)$ since each of the $n$ devices must repeatedly interact with each of the $|\mathcal{C}|$ clusters to determine which cluster to join. Since the number of clusters $|\mathcal{C}|$ may be as large as the number of devices $n$, then hierarchical FAACL has a computational complexity that is at least as good as any other clustered federated learning technique.

## B  Synthetic Data Generation and Data Partition

Here, we provide a breakdown of how synthetic data is created for each setting.

**Asymmetric Image Partition ($A_1$):** The devices are partitioned equally into two clusters. The dataset is split equally and uniformly at random into the devices of the first cluster. The devices of the second cluster receive copies of the data splits as for cluster 1, but each image is perturbed with Gaussian, and salt and pepper noise. This configuration simulates scenarios where data quality or noise levels vary among devices.

**Asymmetric Data Amount Partition ($A_2$):** The devices are partitioned equally into two clusters. Devices in the first cluster receive abundant data while devices in the second cluster receive limited data. The dataset is split equally and uniformly at random into the devices of the first cluster. For the second cluster, one fifth of the dataset is sampled and partitioned equally and uniformly at random among the devices. In addition, for the devices of the second cluster, the labels of digits 1, 2, and 3 are aggregated into a single class with label 2. This partitioning mimics scenarios where devices receive varying amounts of data and varying label granularity.

**Symmetric Label Partition ($S_1$):** The devices are partitioned equally into two clusters. The data with labels 0 to 4 is split equally and uniformly at random into the devices of the first cluster. The data with labels 5 to 9 is split equally and uniformly at random into the devices of the second cluster. This design simulates scenarios where devices have access to data with distinct label ranges.

**Symmetric Predictor Partition ($S_2$):** The devices are partitioned equally into two clusters. The dataset is split equally and uniformly at random into the devices of the first cluster. The devices of the second cluster receive data with shifted labels (i.e., class 0 is relabeled as 1, class 1 is relabeled as 2, ..., class 9 is relabeled as 0). The shifted version of the data set is split equally and uniformly at random into the devices of the second cluster. This partitioning aims to simulate situations where devices have different underlying predictors for the same inputs.

## C  Code, Experimental Parameters and Environment

### C.1  Code

The code for FAACL is available for review in an anonymized github repository: https://github.com/FAACL/FAACL

### C.2  Experimental Setup (Software, Hardware, Randomization)

The implementation was coded in Python. Randomization was done by using three seeds in Numpy. The seeds were set to 10, 55 and 2077 for all the algorithms and datasets. Experiments were run on a single Nvidia GPU (either T4 or A40).

### C.3  Baselines

We evaluate the performance of our algorithm with the following baselines:

- **Centralized FL**: This Oracle method aggregates data from all devices centrally to train a global model. It serves as an idealized benchmark, assuming perfect data availability and no distributional discrepancies.

- **FedAvg** (McMahan et al., 2017): A foundational Federated Learning (FL) approach that updates a global model through the aggregation of local model updates from all participating devices, without employing any clustering strategy.
- **FedGroup** (Duan et al., 2021): A clustered federated learning method that organizes devices into clusters based on the cosine similarity of their gradients.
- **IFCA** (Ghosh et al., 2020): A clustered federated learning approach that dynamically assigns each device to clusters that minimize the local loss.
- **FeSEM** (Xie et al., 2021): A clusters federated learning method that minimize the $l_2$ norm distance between individual device models and their respective cluster models.
- **FedDrift** (Jothimurugesan et al., 2023): Designed to adapt to concept drift in federated continual learning environments, FedDrift assumes initial data homogeneity across devices. For our experiments, we adapt FedDrift to start with one initial cluster per device, implementing its clustering merge strategy iteratively.
- **CFL-GP (Kim et al., 2024)** partitions devices into groups with similar accumulated gradients by spectral clustering.
- **SR-FCA (Vardhan et al., 2024)** successively refines a clustering of devices by bottom up aggregation based on a cross-model loss.

## C.4 DATASETS

We use the following datasets for our experiments:

- **MNIST** (Deng, 2012) A benchmark dataset comprising $28 \times 28$ pixel grayscale images of handwritten digits, categorized into 10 classes.
- **Extended MNIST (EMNIST)** (Cohen et al., 2017) An extension of MNIST to handwritten characters, offering a 62-class image classification challenge. For our experiments, we focus on the first ten characters from 'a' to 'j'.
- **Fashion MNIST (FASHION)** (Xiao et al., 2017) Similar in structure to MNIST, this dataset features $28 \times 28$ pixel grayscale images of fashion items, divided into 10 categories.
- **Federated Extended MNIST (FEMNIST)** (Caldas et al., 2018) A federated learning-specific version of EMNIST, where each device's data originates from a unique writer, featuring a total of 3550 users. We utilize a subset comprising 5% of the data from 197 users.
- **CIFAR10 (Krizhevsky et al., 2009)**: A popular benchark dataset consisting of $32 \times 32$ pixel color images, divided into 10 classes, each representing different objects such as animals and vehicles.
- **Sentiment140 (SENT140)** (Caldas et al., 2018): A federated version of Text Dataset of Tweets. It contains 1,600,000 tweets extracted using Twitter.

An overview of the Datasets and model parameters is shown in Table 5

Table 5: Summary of datasets and models parameters.

| Dataset | Devices | Samples | Parameters |
|---|---|---|---|
| MNIST | 100 | 69,035 | 101,770 |
| EMNIST | 20 | 18,345 | 407,050 |
| FASHION | 50 | 72,505 | 407,050 |
| FEMNIST | 197 | 40,875 | 434,752 |
| CIFAR10 | 100 | 60,000 | 2,074,260 |
| SENT140 | 772 | 40,783 | 232,386 |

## C.5 MODEL ARCHITECTURE

The neural architecture used for dataset MNIST, EMNIST, FASHION, and FEMNIST is the Multi-layer Perceptron, a feedforward neural network with two hidden layers for FEMNIST and one hidden

layer for the other datasets. We also performed L2 regularization and utilized a ReLU activation function and a softmax output layer with a Sparse Categorical Cross-Entropy loss, that is trained using Stochastic Gradient Descent. For the dataset of SENT140, we use the sequential model, starting with an input layer that expects sequences of length 25 with 300 features each. The model uses two bidirectional LSTM (Long Short-Term Memory) layers, which are a type of recurrent neural network (RNN) layer suited for learning from sequences. The first LSTM layer has 64 units and returns sequences, feeding into another bidirectional LSTM layer with 32 units that does not return sequences. This is followed by a dense layer with 64 neurons and ReLU activation, a dropout layer with a rate of 0.5 to prevent overfitting, and finally, a dense output layer with 2 neurons and softmax activation for binary classification.

## C.6 HYPERPARAMETERS

Table 6: Hyperparameter Summary Table for Scenario $S_0, S_1, S_2, A_1, A_2$

| Parameter | Dataset | $S_0$ | $S_1$ | $S_2$ | $A_1$ | $A_2$ |
|---|---|---|---|---|---|---|
| Epochs | MNIST | 300 | 300 | 300 | 300 | 300 |
| | EMNIST | 300 | 300 | 300 | 300 | 300 |
| | FASHION | 300 | 300 | 300 | 300 | 300 |
| | CIFAR10 | 300 | 300 | 300 | 300 | 300 |
| Learning rate | MNIST | 0.01 | 0.01 | 0.01 | 0.01 | 0.01 |
| | EMNIST | 0.003 | 0.003 | 0.003 | 0.003 | 0.003 |
| | FASHION | 0.005 | 0.005 | 0.005 | 0.005 | 0.005 |
| | CIFAR10 | 0.002 | 0.002 | 0.002 | 0.002 | 0.002 |
| $\delta$ | MNIST | 3 | 3 | 3 | 3 | 3 |
| | EMNIST | 2 | 2 | 3 | 3 | 3 |
| | FASHION | 2 | 3 | 3 | 3 | 3 |
| | CIFAR10 | 2 | 3 | 3 | 3 | 3 |
| $\epsilon$ | MNIST | 0.7 | 0.7 | 0.7 | 0.5 | 0.4 |
| | EMNIST | 0.7 | 0.7 | 0.7 | 0.4 | 0.1 |
| | FASHION | 0.7 | 0.7 | 0.7 | 0.5 | 0.1 |
| | CIFAR10 | 0.6 | 0.6 | 0.6 | 0.4 | 0.2 |

In our methodology for identifying potential mergers between clusters $C_1$ and $C_2$, we employ a statistical approach where the significance threshold $\alpha$ is compared against the p-value from a statistical test. This test evaluates the null hypothesis $\ell(C_1.\theta, z) + \epsilon < \ell(C_2.\theta, z)$, aiming to determine the likelihood of a merge based on the model parameters $\theta$ and data point $z$.

For the implementation of FedDrift, we introduce a distance metric $D_{ij}$ representing the proximity between cluster $i$ and cluster $j$. A merge is considered when $D_{ij}$ falls below a predefined threshold $\delta$, indicating a significant overlap in the data representation of both clusters.

Table 7: Hyperparameter Summary Table for Scenario $A_0$

| Parameter | FEMNIST | SENT140 |
|---|---|---|
| Epochs | 300 | 300 |
| Learning rate | 0.005 | 0.005 |
| $\delta$ | 4 | 4 |
| $\epsilon$ | 0.5 | 0.7 |

The number of Epochs, learning rate, $\delta$, and $\epsilon$ are summarized in Table 6 and 7 for different experimental scenarios.

Our experiments incorporate both Gaussian and Salt & Pepper noise to construct the experiments with devices having different data quality. Gaussian noise, characterized by its variance and mean, introduces a continuous perturbation, while salt & pepper noise, specified by a density parameter, simulates random pixel corruptions. The configurations for these noise parameters are outlined in Table 8.

Table 8: Noise Parameters Summary Table

| Parameter | MNIST | EMNIST | FASHION | CIFAR10 |
|---|---|---|---|---|
| Gaussian Noise variance | 0.4 | 1.0 | 0.9 | 0.6 |
| Gaussian Noise mean | 0.0 | 0.0 | 0.0 | 0.0 |
| Salt & Pepper Noise density | 0.7 | 0.6 | 0.7 | 0.4 |

### C.7 ADDITIONAL RESULTS

The table below presents the communication costs observed during the MNIST experiment, which incorporated 20 devices per iteration. This table compares the communication overhead associated with our approach relative to conventional federated learning methods. Although our method initially result in increased communication demands in the first $\log n$ iterations due to forming clusters, he overhead in subsequent iterations reduces to levels comparable to other baselines.

Table 9: Communication Overhead (combined size of all messages between the server and the devices in one communication round) in MNIST Experiments

| | $S_0$ | $S_1$ | $S_2$ | $A_1$ | $A_2$ |
|---|---|---|---|---|---|
| FedAvg | 16.3 MB | 16.3 MB | 16.3 MB | 16.3 MB | 16.3 MB |
| IFCA | 48.9 MB | 48.9 MB | 48.9 MB | 48.9 MB | 48.9 MB |
| FeSEM | 48.9 MB | 48.9 MB | 48.9 MB | 48.9 MB | 48.9 MB |
| FedGroup | 16.3 MB | 16.3 MB | 16.3 MB | 16.3 MB | 16.3 MB |
| FedDrift | 24.5 MB | 73.4 MB | 57.1 MB | 65.2 MB | 32.6 MB |
| FedSoft | 81.5 MB | 81.5 MB | 81.5 MB | 81.5 MB | 81.5 MB |
| CFL-GP | 48.9 MB | 48.9 MB | 48.9 MB | 48.9 MB | 48.9 MB |
| SR-CFA | 16.3 MB | 48.9 MB | 48.9 MB | 36.2 MB | 24.5 MB |
| FAACL(first $\log 20 \approx 5$ rounds) | 70.1 MB | 70.1 MB | 70.1 MB | 101.1 MB | 101.1 MB |
| FAACL(after $\log 20 \approx 5$ rounds) | 16.3 MB | 16.3 MB | 16.3 MB | 20.4 MB | 20.4 MB |

The following table, which compares the performance metrics, including accuracies and execution times, of both the hierarchical and flat versions. This comparison demonstrates the efficiency and effectiveness of the hierarchical approach in achieving the desired clustering results with reduced computational costs.

Table 10: Test accuracies ± stderr [ of clusters for seed 1, of clusters for seed 2, of clusters for seed 3] and execution time (per device in one iteration) comparison between Hierarchical FAACL (H-FAACL) and Flat FAACL (F-FAACL). Standard error for execution time is omitted since it is always less than 1e-5.

| Experiment | $H-FAACL$ accuracy | $H-FAACL$ time | $F-FAACL$ accuracy | $F-FAACL$ time |
|---|---|---|---|---|
| MNIST-$S_0$ | $96.12 \pm 0.99 \, [1, 1, 1]$ | $6.3 \, s$ | $96.32 \pm 0.31 [1, 1, 1]$ | $43.8 \, s$ |
| MNIST-$S_1$ | $93.44 \pm 0.05 \, [2, 2, 2]$ | $4.3 \, s$ | $93.72 \pm 0.18 \, [2, 2, 2]$ | $34.2 \, s$ |
| MNIST-$S_2$ | $95.73 \pm 0.02 \, [2, 2, 2]$ | $3.9 \, s$ | $95.60 \pm 0.63 \, [2, 2, 2]$ | $36.9 \, s$ |
| MNIST-$A_1$ | $75.12 \pm 0.23 \, [4, 4, 4]$ | $8.6 \, s$ | $74.71 \pm 0.83 \, [4, 4, 4]$ | $69.4 \, s$ |
| MNIST-$A_2$ | $96.01 \pm 0.52 \, [5, 5, 5]$ | $10.5 \, s$ | $96.14 \pm 0.39 \, [5, 5, 5]$ | $76.6 \, s$ |
| EMNIST-$S_0$ | $97.09 \pm 0.28 \, [1, 1, 1]$ | $15.2 \, s$ | $96.83 \pm 0.47 \, [1, 1, 1]$ | $174.0 \, s$ |
| EMNIST-$S_1$ | $96.86 \pm 0.11 \, [2, 2, 2]$ | $9.7 \, s$ | $97.06 \pm 0.06 \, [2, 2, 2]$ | $146.7 \, s$ |
| EMNIST-$S_2$ | $96.48 \pm 0.02 \, [2, 2, 2]$ | $8.9 \, s$ | $95.60 \pm 0.63 \, [2, 2, 2]$ | $167.1 \, s$ |
| EMNIST-$A_1$ | $53.08 \pm 0.33 \, [9, 9, 9]$ | $43.1 \, s$ | $52.89 \pm 0.35 \, [9, 9, 9]$ | $783.4 \, s$ |
| EMNIST-$A_2$ | $96.95 \pm 0.18 \, [4, 4, 4]$ | $29.2 \, s$ | $96.42 \pm 0.38 \, [4, 4, 4]$ | $272.2 \, s$ |
| FASHION-$S_0$ | $88.24 \pm 0.44 \, [1, 1, 1]$ | $15.4 \, s$ | $88.31 \pm 0.27 \, [1, 1, 1]$ | $157.0 \, s$ |
| FASHION-$S_1$ | $90.22 \pm 0.18 \, [2, 2, 2]$ | $10.2 \, s$ | $90.10 \pm 0.47 \, [2, 2, 2]$ | $126.7 \, s$ |
| FASHION-$S_2$ | $87.24 \pm 0.08 \, [2, 2, 2]$ | $9.3 \, s$ | $86.89 \pm 0.76 \, [2, 2, 2]$ | $149.3 \, s$ |
| FASHION-$A_1$ | $73.19 \pm 0.06 \, [3, 3, 3]$ | $13.7 \, s$ | $72.85 \pm 0.37 \, [3, 3, 3]$ | $224.9 \, s$ |
| FASHION-$A_2$ | $86.52 \pm 0.01 \, [2, 2, 2]$ | $11.6 \, s$ | $86.63 \pm 0.14 \, [2, 2, 4]$ | $200.3 \, s$ |

Table 11: Test accuracies $\pm$ stderr for $A_1$ with [number of clusters].

| Dataset | MNIST | EMNIST | FASHION | CIFAR10 |
|---|---|---|---|---|
| Centralize | 78.39±0.05[2] | 62.39±0.32[2] | 73.43±0.13[2] | 58.51±0.04[2] |
| Fedavg(optimal) | 77.58±0.11[2] | 60.25±0.13[2] | 73.32±0.20[2] | 57.81±0.07[2] |
| IFCA | 69.90±0.41[5] | 50.50±0.84[5] | 69.09±0.61[4] | 53.29±0.51[5] |
| FeSEM | 65.79±0.82[4] | 46.31±2.24[1] | 64.56±1.32[1] | 48.72±0.92[1] |
| FedGroup | 74.65±0.16[5] | 51.88±0.24[5] | 70.93±0.52[5] | 54.26±0.58[5] |
| FedDrift | 70.34±0.43[7] | 51.63±0.11[4] | 72.96±0.17[3] | 54.58±0.46[1] |
| FedSoft | 71.28±0.31 [5] | 52.38±0.56[5] | 71.01±0.47 [5] | 53.38±0.75[5] |
| CFL-GP | 72.16±0.64 [5] | 52.31±0.56[5] | 70.39±0.60 [5] | 52.14±0.83[5] |
| SR-CFA | 74.27±0.84 [3] | 53.84±0.29[6] | 70.04±0.89 [8] | 51.62±0.59[4] |
| FAACL(version 1) | **76.31±0.13**[4] | **57.28±0.58**[9] | **73.21±0.10**[3] | **56.31±0.48**[6] |
| FAACL(version 2) | 75.12±0.34[4] | 55.85±0.43[3] | 73.19±0.32[3] | 55.42±0.68[4] |

Table 12: Test accuracies $\pm$ stderr for $A_2$ with [number of clusters].

| Dataset | MNIST | EMNIST | FASHION | CIFAR10 |
|---|---|---|---|---|
| Centralize | 97.32±0.03[2] | 97.67±0.29[2] | 86.90±0.24[2] | 64.16±0.09[2] |
| Fedavg(optimal) | 97.10±0.17[2] | 97.38±0.25[2] | 86.72±0.14[2] | 64.04±0.08[2] |
| IFCA | 95.35±0.28[3] | 95.98±0.18[2] | 85.70±0.32[4] | 59.17±0.62[5] |
| FeSEM | 74.65±0.20[1] | 88.02±1.27[1] | 81.27±0.30[1] | 52.84±1.28[5] |
| FedGroup | 95.44±0.26[5] | 95.42±0.18[5] | 86.31±0.28[5] | 60.27±0.61[5] |
| FedDrift | 95.34±0.16[3] | 91.70±0.45[4] | 84.15±0.35[2] | 60.28±1.14[8] |
| FedSoft | 95.73±0.09 [5] | 95.73±0.09 [5] | 84.29±0.58 [5] | 60.25±0.79[5] |
| CFL-GP | 94.36±0.36 [5] | 94.24±0.37[5] | 83.25±0.60 [5] | 61.17±1.02[5] |
| SR-CFA | 95.38±0.25 [2] | 96.48±0.92[5] | 84.57±0.29 [5] | 61.09±0.77[5] |
| FAACL(version 1) | **96.40±0.37**[5] | 96.95±0.23[4] | **86.53±0.14**[2] | **62.74±0.43**[8] |
| FAACL(version 2) | 96.06±0.26[4] | **97.02±0.39**[3] | 86.50±0.07[2] | 62.43±0.52[4] |

Table 13: Test accuracies $\pm$ stderr for $S_1$ with [number of clusters].

| Dataset | MNIST | EMNIST | FASHION | CIFAR10 |
|---|---|---|---|---|
| Centralize | 94.85±0.16[2] | 97.11±0.09[2] | 90.82±0.12[2] | 74.68±0.11[2] |
| FedAvg(optimal) | 94.39±0.18[2] | 97.09±0.06[2] | 90.47±0.14[2] | 74.50±0.26[2] |
| IFCA | 91.16±0.38[5] | 94.28±0.46[4] | 86.88±0.24[4] | 69.36±0.31[5] |
| FeSEM | 50.24±3.80[3] | 42.44±3.88[1] | 50.57±1.55[1] | 54.20±0.71[1] |
| FedGroup | **93.73±0.13**[5] | 96.18±0.14[5] | 88.50±0.44[5] | **71.62±0.29**[5] |
| FedDrift | 91.76±0.11[8] | 96.35±0.20[3] | 85.52±0.24[7] | 70.53±0.79[3] |
| FedSoft | 90.49±0.25 [5] | 94.39±0.71 [5] | 84.29±0.36 [5] | 72.49±0.33[5] |
| CFL-GP | 92.31±0.48 [5] | 96.73±0.65[5] | 88.29±0.58 [5] | 70.35±0.64[5] |
| SR-CFA | 92.16±0.24 [2] | 96.01±0.43[2] | 89.26±0.79 [2] | 70.23±0.71[2] |
| FAACL(version 1) | 93.45±0.03[2] | **96.82±0.13**[2] | **90.23±0.11**[2] | 71.48±0.27[2] |
| FAACL(version 2) | 93.27±0.06[2] | 96.44±0.21[2] | 89.64±0.22[2] | 71.27±0.23[2] |

Table 14: Test accuracies $\pm$ stderr for $S_2$ with [number of clusters].

| Dataset | MNIST | EMNIST | FASHION | CIFAR10 |
|---|---|---|---|---|
| Centralize | 97.06±0.04[2] | 96.93±0.13[2] | 88.95±0.68[2] | 73.57±0.11[2] |
| FedAvg (optimal) | 96.73±0.29[2] | 96.80±0.37[2] | 88.74±0.49[2] | 73.18±0.22[2] |
| IFCA | 94.36±0.52[4] | 95.17±0.05[2] | 85.42±0.48[5] | 71.11±0.23[4] |
| FeSEM | 49.35±4.17[3] | 43.94±3.05[1] | 43.65±1.65[1] | 64.76±1.32[3] |
| FedGroup | 95.55±0.22[5] | 95.79±0.22[5] | 85.98±0.09[5] | 70.81±0.41[5] |
| FedDrift | 93.37±0.35[6] | 96.12±0.17[3] | 85.77±0.28[4] | 71.30±0.55[3] |
| FedSoft | 93.92±0.41[5] | 94.78±0.63[5] | 85.11±0.26[5] | **71.58±0.57**[5] |
| CFL-GP | 94.92±0.61[5] | 95.75±0.82[5] | 86.02±0.73[5] | 71.21±0.88[5] |
| SR-CFA | 95.06±0.73[2] | 95.88±0.46[2] | 86.69±0.40[2] | 71.47±0.64[2] |
| FAACL(version 1) | **95.89±0.11**[2] | 96.54±0.07[2] | **87.82±0.17**[2] | 71.47±0.27[2] |
| FAACL(version 2) | 95.71±0.04[2] | **96.62±0.09**[2] | 87.31±0.13[2] | 71.43±0.29[2] |

