# OpenReview forum: "FAACL: Federated Adaptive Asymmetric Clustered Learning"
_ICLR.cc/2025/Conference — Submitted to ICLR 2025_

### Official Review · Reviewer_WX7G · 2024-11-02

**Soundness:** 2
**Presentation:** 1
**Contribution:** 2
**Rating:** 3
**Confidence:** 5

**Summary:**

This paper presents a clustered federated learning method named FAACL, which aims to adaptively cluster devices and dispatch models for each device according to its cluster.

**Strengths:**

The author provides a theoretical analysis of its proposed cluster strategy.

**Weaknesses:**

This paper seems to be an incremental work of CFL. The contribution is limited.

This paper is poorly written. E.g., 1) the author should add a figure of framework and workflow. 2) The expression of lines 186-196 seems to detail the contributions and steps of the proposed method, rather than the components. I advise the author to reorganize the expressions.

The author attempts to utilize CFL-based strategy to address the problem of non-IID data. However, the author lacks comparison the proposed method with SOTA tradition federated learning optimization methods [1-4].

The proposed method cannot adapt to the secure aggregation method. The author should analyze the privacy of the proposed method.

The experiments are mainly based on MNIST and its variants. I would like to know how the method performs on more complex datasets, such as ImageNet. Please note that the experimental results show that the proposed method is not effective enough on the cifar10 dataset and is even inferior to the baseline.

**Questions:**

Please see the weakness.

---

> ### Author Response · Authors · 2024-11-22
> **clarification**
>
> Thank you for the feedback.  We have a clarification question.  Above you indicated that our method lacks a comparison to SOTA techniques [1-4].  Can you clarify what are the SOTA techniques [1-4] that you are referring to.
>
> Note that we are currently revising the paper to address each point raised and we will post the revisions once completed.

---

> > ### Comment · Reviewer_WX7G · 2024-11-22
> >
> > I am so sorry for the mistake. The related methods are as follows:
> >
> > [1] Generalized Federated Learning via Sharpness Aware Minimization, ICML 2022
> >
> > [2] Is Aggregation the Only Choice? Federated Learning via Layer-wise Model Recombination, KDD 2024
> >
> > [3] FedCross: Towards Accurate Federated Learning via Multi-Model Cross-Aggregation, ICDE 2024
> >
> > [4] Clustered Federated Learning via Gradient-based Partitioning. ICML 2024

---

> > > ### Author Response · Authors · 2024-11-28
> > > **rebuttal (part 1)**
> > >
> > > Thank you for the critical feedback.  Please find some answers to your questions below.
> > >
> > > **Incremental work**:  Note that none of the existing CFL techniques consider asymmetric clustering.  We are the first to describe the limitations of symmetric clustering and to propose an asymmetric clustering technique.  This is entirely novel and distinct from the CFL literature that considers only symmetric clustering.
> > >
> > > **Workflow figure**:  Thank you for the suggestion to add a figure that summarizes the workflow of FAACL.  In the original paper we included several small figures for each step of FAACL, but we agree that this does not give the overall picture of the proposed approach.  Hence, we replaced all the small figures by one overall flowchart in Figure 1.
> > >
> > > **Lines 186-196**:  We apologize for the confusion of those lines.  We rewrote this text (now in Lines 188-192) to simply introduce the different parts of FAACL and their associated sections.
> > >
> > > **SOTA comparison**:  Thank you for pointing out 4 related papers.  We added an empirical comparison to CFL-GP since it is a CFL technique.  The results for CLF-GP are highlighted in blue in each table.  The other 3 papers focus on sharpness aware optimization to find a single global model in a flat area for better generalization.  They solve an orthogonal problem to CFL.  CFL enables distributed learning in scenarios where different clients require different models.  More precisely, when the underlying conditional class distribution P(y|x) is significantly different for different clients, then there does not exist any global classifier that achieves reasonable accuracy for all clients simultaneously.  In contrast, sharpness aware techniques assume mild differences between the underlying conditional class distributions P(y|x) of each client and therefore the existence of a single classifier that can generalize well and achieve good accuracy for all clients.  These techniques do use multiple intermediate models to escape local optima and improve convergence to flat areas, but this is very different than the multiple models in CFL. Ultimately, sharpness aware techniques combine the intermediate models into a single final model for all clients, while CFL techniques return different final models for each cluster because the underlying class conditional distribution is too different for any single classifier to be accurate for all clients (regardless of any flat region).
> > >
> > > **Privacy**: We added Section 4.6 in the revised paper to explain how to enhance the privacy of FAACL with differential privacy (DP).  FAACL can be combined with a local DP mechanism (Shokri
> > > & Shmatikov, 2015) by adding noise to the weights of each model before sharing. In Algorithm 6,
> > > each device can add Gaussian noise as a function of sensitivity to achieve a desired degree of privacy
> > > at the end of each round of training. Models can then be shared with clusters and other devices while
> > > statistically preventing membership attacks at the cost of a reduction in accuracy.  Note that secure aggregation is insufficient to ensure privacy.  Secure aggregation keeps each local model private, but the aggregated model remains visible to everyone and can be attacked to infer the underlying data, especially when the aggregated model memorizes the data.  In contrast, differential privacy (statistically) prevents membership attacks.

---

> > > > ### Author Response · Authors · 2024-11-28
> > > > **rebuttal (part 2)**
> > > >
> > > > **Additional dataset**:  We realize that it is always desirable to do more experiments on more datasets, but we did not conduct the requested experiment on ImageNet for several reasons. First, it is not clear what hypothesis this experiment would verify.  As you pointed out, ImageNet is more complex than CIFAR10, but this simply means that it is more challenging for computer vision as opposed to federated learning.  In fact, ImageNet is not a dataset designed to test federated learning algorithms since there is no notion of clients that possess different distributions of images.  We could split ImageNet into different clients to create a synthetic task, but we already did that with 4 other datasets.  We also reported results on two natural FL datasets.  Note also that none of the CFL baselines (FedGroup, IFCA, FeSEM, FedDrift, CFL-GP, SR-FCA) were tested on ImageNet in their respective paper. Furthermore, our empirical evaluation includes more datasets (6) than the empirical evaluations in the respective papers of the CFL baselines (FedGroup (4 datasets), IFCA (4 datasets), FeSEM (3 datasets), FedDrift (5 datasets), CFL-GP (5 datasets), SR-FCA (4 datasets)).
> > > >
> > > > **CIFAR10 performance**: FAACL did not achieve the best performance on MNIST and CIFAR10 in Tables 3 and 4.  Note that Tables 3 and 4 report results for the symmetric data partitions, which means that asymmetric clustering (FAACL's main algorithmic advantage) plays no role. Hence, it is normal for FAACL not to win on every dataset in the symmetric settings.  In fact, the goal of the symmetric settings is to demonstrate that FAACL remains competitive (i.e., there is no tradeoff leading to a performance loss due to the use of asymmetric clustering in settings without any asymmetry).  FAACL still scores among the best techniques on MNIST and CIFAR10 and it reaches an accuracy within 1-3% of the oracle upper bound provided by FedAvg (optimal) which has access to the ground truth clustering.  Ultimately, the experiments that matter the most are those on the real FL datasets, FEMNIST and SENT140, where FAACL improved the state of the art and found asymmetries that it leveraged to boost performance.
> > > >
> > > > If the above answers address your concerns, please consider revising your score.  Otherwise, let us know of any outstanding concern.

---

### Official Review · Reviewer_7bx5 · 2024-11-04

**Soundness:** 2
**Presentation:** 3
**Contribution:** 2
**Rating:** 3
**Confidence:** 3

**Summary:**

This paper proposes a new federated learning technique where some devices may contribute to the training of the models of other devices, but without enforcing reciprocity, leading to a form of asymmetric clustering. This approach not only enhances data utilization across the devices, but also maintains the integrity of high-quality data.

**Strengths:**

1. The proposed approach tackles the underexplored issue of asymmetric clustering in Clustered Federated Learning, providing a fresh perspective.

2. The experimental results are comprehensive.

**Weaknesses:**

1. The paper presents a clustered federated learning method; however, it lacks an introduction to the implementation of federated learning (FL) and clustered federated learning (CFL). I recommend including details about the specific training processes and objectives of both FL and CFL in the BACKGROUND section to enhance clarity and context.

2. How can the proposed approach be effectively implemented in real-world scenarios? In practical federated learning settings, it seems challenging to determine whether cluster C2 supports cluster C1. Given that devices do not have visibility into each other's data during training, the primary method for assessing support appears to require multiple rounds of training and testing. This approach could lead to considerable resource inefficiencies.

3. The main operations of the method are presented in algorithmic form in the appendix (e.g., Algorithms 5 and 6). However, it may be more beneficial for readers to understand these concepts if they are described in text form within the main body of the paper.

4. In the experimental setup, there is no detailed information on how the data is divided or the rationale for this division. Different data partitioning methods can impact the final performance of federated learning.

**Questions:**

See weaknesses

---

> ### Author Response · Authors · 2024-11-28
> **rebuttal**
>
> Thank you for the constructive feedback.  Please find some answers to your questions below.
>
> 1.  We added an introduction to federated learning and clustered federated learning in blue in Section 2.
>
> 2. FAACL can be implemented as it is for real world scenarios.  In fact, this work was done in collaboration with an industry partner.  To assess support, devices do not send their data to other devices. Instead, they receive the models of other clusters and evaluate them on their private validation set.  Training is done in multiple rounds as in most federated learning techniques.  FAACL and all the baselines used in the empirical comparison use FedAvg as the internal subroutine to train each cluster model.  To ensure convergence, FedAvg requires multiple rounds of training.  Perhaps the concern is that support evaluation will require additional rounds of training and testing.  Note that in all clustered federated learning techniques additional rounds are needed to determine the cluster membership of each device and to continue the training when cluster membership changes.  Here cluster support evaluation is equivalent to (though conceptually and procedurally different than) cluster membership evaluation in other CFL techniques.  Table 9 compares the communication cost (i.e., combined size of all communication messages) of FAACL and the other CFL baselines.  It is higher in the first 5 rounds since FAACL learns the number of clusters  and then becomes similar to the other CFL baselines in subsequent rounds.  Note that all algorithms ran for the same number of rounds (300) as listed in Tables 6, 7 (rounds correspond to epochs in Tables 6, 7).
>
> 3. We added some text in blue in Lines 249-252 to provide more details about Algorithm 5 and in Lines 257-260 for Algorithm 6.
>
> 4. Regarding the data splits, we are not sure whether you missed the text in Appendix B or you saw Appendix B but didn't find it clear enough.  In any case, we added more details to Appendix B.  Let us know if you still have questions about the data splits.
>
> If the above answers address your concerns, please consider revising your score.  Otherwise, let us know of any outstanding concern.

---

### Official Review · Reviewer_twgm · 2024-11-04

**Soundness:** 2
**Presentation:** 3
**Contribution:** 2
**Rating:** 5
**Confidence:** 2

**Summary:**

Authors discuss the clustered learning approach in Federated Learning and relax the following constraints:
1. Number of clusters being fixed (Adaptive)
2. Each device contributes to a single cluster (Adaptive)
3. Model sharing is reciprocal in nature (Asymmetric)

**Strengths:**

S1: Introduced asymmetric clustering
S2: Source code is shared for reproducibility.

**Weaknesses:**

W1: This comment is in relation to the Section 4.2 and 4.3 -> Some of the related works may need to be studied and probably be included as part of the comparison metrics. [1-2]

W2: How does this handle a newly added device in the Federated Learning? Do we check against all the clusters and find if it supports or not individually or against the clusters only? Does that a performance hit if the clusters are already prepared.

[1] Vardhan, Harsh, Avishek Ghosh, and Arya Mazumdar. "An Improved Federated Clustering Algorithm with Model-based Clustering." Transactions on Machine Learning Research (2024).

[2] Kim, Heasung, Hyeji Kim, and Gustavo De Veciana. "Clustered Federated Learning via Gradient-based Partitioning." Forty-first International Conference on Machine Learning.

**Questions:**

Q1 :  line 053 - “and all existing techniques assume that each device contributes to a single cluster.” - What does this line mean here? And how do address the same in your paper?

Q2: Does it have the same privacy guarantees as other models proposed in the same vein such as [1] and others.

Q3: How does the computational complexity compare to other algorithms?

Q4: How far off is this clustering from ground truth? Additionally is there a comparison again Oracle? [2]

Q5: Was there an ablation study performed without merging of clusters and just having asymmetric contribution?

[1] Sattler, Felix, Klaus-Robert Müller, and Wojciech Samek. "Clustered federated learning: Model-agnostic distributed multitask optimization under privacy constraints." IEEE transactions on neural networks and learning systems 32.8 (2020): 3710-3722.

[2] Jothimurugesan, Ellango, et al. "Federated learning under distributed concept drift." International Conference on Artificial Intelligence and Statistics. PMLR, 2023.

---

> ### Author Response · Authors · 2024-11-28
> **rebuttal (part 1)**
>
> Thank you for the constructive feedback.  Please find some answers to your questions below.
>
> **W1**: Thank you for pointing those two very recent papers on CFL.  We added a description of CFL-GP and SR-FCA in the related work section and included CFL-GP and SR-FCA in the empirical evaluation.  The results for those techniques are highlighted in blue in each table of the revised paper.  In short, FAACL outperforms CFL-GP and SR-FCA in all settings except for the MNIST dataset in Table 3 where the differences are negligible given that they both achieve an accuracy similar to the upper bound provided by FedAvg (optimal) when given the true underlying optimal cluster.
>
> **W2**: FAACL can naturally handle newly added devices.  We added the following text in blue in Lines 316-321: "In practice, devices may enter and leave the federation at any time. When a new device appears, it is
> simply initialized as a singleton cluster whose model is the local model of that device. Then this new
> cluster participates in subsequent iterations of cluster merging and cluster updating as usual. When a
> device leaves the federation, it is simply removed from the support and membership of each cluster it
> used to contribute to. If this device was part of a singleton cluster, that cluster is deleted. Subsequent
> iterations of cluster merging and cluster updating proceed as usual."
>
> **Q1**: In previous work, when it is assumed that each device contributes to a single cluster, this limiting assumption prevents devices from contributing to the training of other clusters, which may lead to suboptimal results. As explained in Lines 66-76, there are asymmetric scenarios where the benefits of model sharing are not
> reciprocal between devices. For instance, consider a situation involving two devices, device $A$ and
> device $B$. Device $A$ has a large dataset characterized by the underlying conditional class distribution
> $p_A(y|x)$, whereas device $B$ has a smaller dataset with a similar conditional class distribution $p_B(y|x)$
> that matches $p_A(y|x)$ for 90% of the inputs $x$.  For device $A$, incorporating device $B$’s data could
> potentially introduce a bias that might degrade the accuracy of its own model because of the 10%
> divergence in their data distributions. Thus device $A$ would not wish to train on data from device
> $B$. Hence $A$ and $B$ should be in different clusters.  On the other hand, having $A$ help $B$ train its model could significantly reduce variance owing to the greater volume of data $B$ would benefit from, thereby enhancing its overall performance (reduction in variance outweighs the bias introduced).  In such a situation, it would be beneficial to let device $A$ contribute to the training of the models of both clusters that A and B belong to.  In FAACL, this is achieved by introducing the notion of support cluster as described in Lines 132-146.  Each cluster model will be trained by the devices of the cluster, but also the devices of supporting clusters (see Algorithm 6).
>
> **Q2**: Regarding privacy, we added Section 4.6 in the revised paper to explain how to enhance the privacy of FAACL with differential privacy (DP).  FAACL can be combined with a local DP mechanism (Shokri
> & Shmatikov, 2015) by adding noise to the weights of each model before sharing. In Algorithm 6,
> each device can add Gaussian noise as a function of sensitivity to achieve a desired degree of privacy
> at the end of each round of training. Models can then be shared with clusters and other devices while
> statistically preventing membership attacks at the cost of a reduction in accuracy.  Note that secure aggregation as suggested by Sattler et al. (2020) is insufficient to ensure privacy.  Secure aggregation keeps each local model private, but the aggregated model remains visible to everyone and can be attacked to infer the underlying data, especially when the aggregated model memorizes the data.  In contrast, differential privacy (statistically) prevents membership attacks.
>
> **Q3**: The computational complexity of Hierarchical FAACL is at least as good as any other CFL technique.  In Lines 919-923, we added the following text in blue: "In
> comparison, all clustered federated learning techniques have a complexity of least $O(n|\mathcal{C}|)$ since
> each of the $n$ devices must repeatedly interact with each of the $|\mathcal{C}|$ clusters to determine which cluster to join. Since the number of clusters $|\mathcal{C}|$ may be as large as the number of devices $n$, then hierarchical FAACL has a computational complexity that is at least as good as any other clustered federated learning technique."

---

> > ### Author Response · Authors · 2024-11-28
> > **rebuttal (part 2)**
> >
> > **Q4**: In each table, we report the number of clusters found by each technique in square brackets.
> >  As explained in Lines 459-463, the FedAvg (optimal) baseline is given the true underlying optimal clustering and therefore the number in brackets for FedAvg (optimal) is the correct number of clusters.  When a technique finds a clustering of the same size then it means that it has found the correct number of clusters.  Furthermore, since all techniques use FedAvg internally to aggregate models in a cluster, the difference in accuracy between a technique and FedAvg (optimal) corresponds to the loss due to a suboptimal clustering.  Finally, note that Theorem 4.1 lower bounds the probability that FAACL will cluster correctly devices with the same data distribution.
> >
> >  **Q5**: We did not do an ablation study for the FAACL variant that omits cluster merging due to the prohibitively high computational cost.  Note that without cluster merging, each device will remain a singleton cluster and the model of each device will still be trained by all relevant supporting devices.  Ultimately, the set of supporting devices for each model will contain the devices that would otherwise be part of a cluster as well as the remaining supporting devices.  Hence, each cluster model would be effectively duplicated as many times as the size of the cluster and training will be multiplied by the same amount.  Accuracy is expected to be similar to FAACL with cluster merging, but with an important blow up in running time.  The time saved by avoiding the merging procedure (Algorithm 5) is negligible compared to the duplication of model training (Algorithm 6) and support cluster evaluation (Algorithms 3 and 4).
> >
> > If the above answers address your concerns, please consider revising your score.  Otherwise, let us know of any outstanding concern.

---

> > > ### Comment · Reviewer_twgm · 2024-12-02
> > > **Some more queries**
> > >
> > > W1 - The datasets both the papers use are different from yours? I am not able to see the similarity in that, and correspondingly similarity on the results as well when compared with the tables presented on the papers.
> > >
> > > For EMNIST, [your algo] 97.31 +- 0.72 with [CFL] 97.29 +- 0.35, if we go ahead with some sanity, now probably your algorithm only performs better in FASHION dataset in Table 3. This may be explained.
> > >
> > > Q2 - The newly added paragraph or subsection in 4.6 does not add anything of value to the paper. Question may have been  tackled in the context of your contribution.

---

> > > > ### Author Response · Authors · 2024-12-04
> > > > **response to additional queries (part 1)**
> > > >
> > > > Thank you for the additional queries.  Please find below some answers to the queries.
> > > >
> > > > **"W1 - The datasets both the papers use are different from yours? I am not able to see the similarity in that, and correspondingly similarity on the results as well when compared with the tables presented on the papers."**
> > > >
> > > > Thank you for raising this point.  Note that this is a problem that predates our work. Different papers use different neural architectures for the same algorithms and different data partitions, different number of devices and different number of data samples per device for the same datasets.  Hence, the results for the same algorithm-dataset pairs necessarily vary across papers due to those differences.  When we started our experiments we wanted to ensure consistency with previous settings, but this was already impossible.  In the end, we elected to use the same multi-layer perceptron architecture as FedGroup since it was the best of the two neural network architectures that they tested.  Since none of the previous works tested asymmetric clustering, we ended up creating new data partitions to evaluate the effectiveness of asymmetric clustering in Table 2.  FEMINST is a natural federated learning dataset that does not require any synthetic data partition, but there are still different versions of this dataset out there.  For instance, Caldas et al. (2018) first created the FEMNIST based on the earlier EMNIST dataset, but FedGroup introduced another dataset based on EMNIST that they also called FEMNIST instead of reusing the original FEMNIST dataset created by Caldas et al. (2018). We are using the original FEMNIST dataset created Caldas et al. (2018).  Please see Appendix B for a detailed description of the data partitions that we used for each dataset, Appendix C.4 for the number of devices and the amount of data per device for each dataset, and Appendix C.5 for a description of the common neural network architecture that we used to ensure fairness across algorithms.
> > > >
> > > > Caldas, and Duddu, Wu, Li, Konecny, McMahan, Smith, Talwalkar (2018) Leaf: A benchmark for federated settings, arXiv:1812.01097.
> > > >
> > > > **"For EMNIST, [your algo] 97.31 +- 0.72 with [CFL] 97.29 +- 0.35, if we go ahead with some sanity, now probably your algorithm only performs better in FASHION dataset in Table 3. This may be explained."**
> > > >
> > > > We agree that the performance differences between FAACL and the other algorithms are not significant in Table 3.  This is expected and as mentioned in the paper the goal was simply to demonstrate that FAACL remains competitive in symmetric settings (without any asymmetry).  Let us clarify the overall goal of each experiment.  The first experiment in Table 1 shows that FAACL improves the state of the art on natural federated learning datasets.  FAACL created asymmetric clusterings and outperformed all baselines since they do not deal with asymmetries.  Since we do not know what are the true clusterings and the degree of asymmetry in those natural datasets, we carried out additional synthetic experiments reported in Table 2 where we explicitly created data partitions with asymmetries.  This confirmed that FAACL handles asymmetries adequately, but not the baselines.  One could ask whether FAACL's ability to deal with asymmetries comes at the cost of a reduction in performance in symmetric settings (without any asymmetry).  The results in Tables 3 and 4 confirm that there is no loss since FAACL remains competitive with the best baselines in symmetric settings (without any asymmetry).  Hence the fact that FAACL achieves results similar to the best performing baselines in Tables 3 and 4 is nothing negative, but simply what is expected given that the goal was to demonstrate that there is no tradeoff induced by attempting to deal with asymmetries when there are none.

---

> > > > > ### Author Response · Authors · 2024-12-04
> > > > > **response to additional queries (part 2)**
> > > > >
> > > > > **"Q2 - The newly added paragraph or subsection in 4.6 does not add anything of value to the paper. Question may have been tackled in the context of your contribution."**
> > > > >
> > > > > The privacy subsection that we added makes a concrete suggestion about how to combine FAACL with differential privacy (DP).  There is no point in devising a new DP technique specifically for FAACL when existing DP techniques can be combined with FAACL.  If the concern is that we did not implement the proposed combination, this is simply because this would be orthogonal to the contribution of the paper (asymmetric clustering).  Note also that none of the CFL baselines (IFCA, FeSEM, FedGroup, FedDrift, FedSoft, CFL-GP, SR-FCA) proposed novel encryption or DP techniques since this would be orthogonal to their contribution.  In fact, IFCA, FeSEM, FedGroup, FedSoft and SR-FCA do not mention anything about enhancing privacy by secure aggregation or DP.  The authors of CFL-GP simply write "the authors recommend the integration of additional privacy safeguards in privacy-sensitive scenarios, e.g., the use of encryption methods and differential privacy techniques" without proposing anything concrete.  The authors of FedDrift simply write "our methods could be combined with other privacy-preserving techniques, e.g., model perturbation (Kairouz et al., 2021, §4) in future work" without more details.  Finally, Sattler et al. (2020) proposed a concrete way to combine their CFL technique with a secure aggregation technique, but did not implement it and secure aggregation is insufficient to ensure privacy (as noted earlier).  Similarly, the subsection that we added provides a concrete suggestion for combining FAACL with a DP technique.  Again, there is no need for reinventing the wheel when existing DP techniques can be combined with FAACL and this is orthogonal to the contribution of our paper.

---

### Author Response · Authors · 2024-12-04
**General comments**

As the discussion period with the authors comes to a close, we'd like to draw the attention of the AC and the reviewers to a few points.  First, given the current low scores of the paper, it will be tempting to simply discard the paper as hopeless.  However, the current scores probably do not take into account the revised version that we submitted and our point-by-point response since there was no feedback from two of the reviewers and only one reviewer posted additional queries on the last day of the discussion period.  Please do read the revised version and each response in the rebuttal before making a decision.

After receiving the reviews we did ponder about what to do with the paper given the low scores.  Since there was no fundamental concern about FAACL and all the points raised were addressable, we revised the paper and provided a response to each point.  We truly appreciate all the constructive feedback, which really helped to improve the paper.  As far as we are concerned, we believe that we have addressed satisfactorily all points.  We took the feedback seriously and added results for two new baselines as requested.  We also revised several sections of the paper as requested.  All material changes are highlighted in blue in the revised paper.

In case the following is not well understood, it is important to note that ICLR is different from other conferences because of the opportunity to revise the paper as part of the rebuttal period.  In other conferences, feedback that requires changes to a paper cannot be addressed, but with ICLR it is possible.  Again, we believe we have adequately addressed all points, but naturally you shall be the judge of that.  If the paper is still rejected, please make sure to include new feedback that we can act on.  We would like to make sure that whatever we re-submit to the next conference does not end up being the same as the revised version that we already uploaded to ICLR due to a lack of new feedback.

---

### Meta-Review · Area_Chair_aK8R · 2024-12-20

**Metareview:**

This paper proposes to create an asymmetric version of clustering in federated learning (training), where the clubbing of different compute nodes may not be bidirectional.

The review scores were pretty low with all recommending rejection. One point that goes against the paper is to further complicate an already involved system while the improvements do not fully compensating for this.

Overall, I recommend rejection.

**Additional Comments On Reviewer Discussion:**

There were not much discussion as review scores were pretty low with some fundamental objections about modeling.

---

### Decision · Program_Chairs · 2025-01-22

Reject